# Bongard in Wonderland:
# Visual Puzzles that Still Make AI Go Mad?

**Antonia Wüst** [1]  **Tim Woydt** [1]  **Lukas Helff** [1 2]  **Inga Ibs** [3 4]  **Wolfgang Stammer** [1 2]  **Devendra S. Dhami** [5]
**Constantin A. Rothkopf** [2 3 4]  **Kristian Kersting** [1 2 4 6]

## Abstract

Recently, newly developed Vision-Language Models (VLMs), such as OpenAI's o1, have emerged, seemingly demonstrating advanced reasoning capabilities across text and image modalities. However, the depth of these advances in language-guided perception and abstract reasoning remains underexplored, and it is unclear whether these models can truly live up to their ambitious promises. To assess the progress and identify shortcomings, we enter the wonderland of Bongard problems, a set of classic visual reasoning puzzles that require human-like abilities of pattern recognition and abstract reasoning. With our extensive evaluation setup, we show that while VLMs occasionally succeed in identifying discriminative concepts and solving some of the problems, they frequently falter. Surprisingly, even elementary concepts that may seem trivial to humans, such as simple spirals, pose significant challenges. Moreover, when explicitly asked to recognize ground truth concepts, they continue to falter, suggesting not only a lack of understanding of these elementary visual concepts but also an inability to generalize to unseen concepts. We compare the results of VLMs to human performance and observe that a significant gap remains between human visual reasoning capabilities and machine cognition. [1]

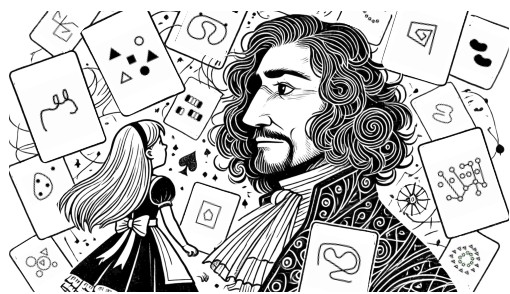

Figure 1: **The wonderland of Bongard problems.** The challenging puzzles circle around simple black-and-white diagrams that picture geometric shapes, patterns, or lines arranged in various ways. Despite their simple appearance, the underlying concepts can be abstract and complex.

## 1. Introduction

Visual reasoning, the ability to understand, interpret, and reason about the visual world, is a fundamental aspect of human intelligence (Marr, 2010). It allows us to navigate our environment, interact with objects, and make sense of complex visual scenes. In recent years, the field of artificial intelligence (AI) has advanced rapidly toward replicating aspects of this visual reasoning, with significant focus placed on Vision-Language Models (VLMs) (Bordes et al., 2024; Liu et al., 2023; 2024a). These models integrate visual and textual information to generate descriptive content, aiming to mimic how humans comprehend and reason about the physical world. Because of their human-like responses, VLMs often create the illusion of possessing human-like perception and intelligence. However, as recent work shows, VLMs and the Large Language Models (LLM) on which they are based have dramatic shortcomings in the case of reasoning (Nezhurina et al., 2024) and visual perception (Kamath et al., 2023; Rahmanzadehgervi et al., 2024; Geigle et al., 2024; Gou et al., 2024) or their combination (Zhou et al., 2023; Zhang et al., 2024; Wang et al., 2024b).

Bongard problems (BPs), a class of visual puzzles that require identifying underlying rules based on a limited set of images, provide a unique and challenging benchmark for assessing visual reasoning abilities in AI (Bongard & Hawkins,

---

[1]AIML Lab, TU Darmstadt [2]Hessian Center for AI (hessian.ai) [3]Institute of Psychology, TU Darmstadt [4]Centre for Cognitive Science, TU Darmstadt [5]Uncertainty in AI Group, TU Eindhoven [6]German Center for AI (DFKI). Correspondence to: Antonia Wüst .

[1]Our code is available at https://github.com/ml-research/bongard-in-wonderland.

1970). Conceived by Mikhail Bongard in 1970, these visual puzzles test cognitive abilities in pattern recognition and abstract reasoning, posing a formidable challenge even to advanced AI systems (Hernández-Orallo et al., 2016).

A BP consists of twelve diagrams, divided into two sides. For each side, a distinct and specific conceptual theme must be identified, which clearly differentiates it from the other side. Although the diagrams themselves are visually simple (see Figure 1), the underlying concepts that connect the images within each group can be abstract, such as *more filled objects than outlined objects* or *turning direction of spiral shape*. Thus, unlike pattern recognition in classification tasks, BPs are not about finding visual patterns in a single diagram that match a certain concept but about finding concepts that allow for the description of a set of diagrams.

While traditional machine learning approaches have made some early progress on BPs (Raghuraman et al., 2023; Depeweg et al., 2024), the potential of VLMs remains largely unexplored. Given that VLMs already struggle with recognizing relatively simple visual patterns (Rahmanzadehgervi et al., 2024; Zhang et al., 2024), BPs present a particularly challenging task, offering a valuable framework for investigating which patterns are easier or harder for state-of-the-art models to identify. Recent work has explored BPs in the context of VLM evaluation (Małkiński et al., 2025). While their study provides meaningful insights, it does not analyze model behavior and failures in greater depth. This highlights the need for a more comprehensive investigation into how VLMs process and reason about BPs.

In this work, we assess the performance of VLMs in solving Bongard problems (BPs) across multiple task settings. (1) We evaluate how effectively different VLMs can identify the underlying rules in BPs through both an open-ended problem-solving approach and a multiple-choice format. (2) We then compare their performance to the results of a newly conducted human study, offering insights into how VLMs measure up against human reasoning. (3) Next, we explore the models' pattern recognition abilities in greater detail by testing their capacity to classify images based on BP rules. (4) Finally, we examine their ability to generate correct hypotheses for the given problems. Our findings provide valuable insights into the perceptual madness of VLMs and suggest opportunities for improvement.

## 2. Related Work

**Bongard and ML.** Depeweg et al. (2024) define a formal language to represent compositional visual concepts. Using this language and Bayesian inference, concepts can be induced from the examples provided in each problem. For a subset of 35 problems, there is reasonable agreement between the concepts with high posterior probability and the solutions formulated by Bongard himself. Raghuraman et al. (2023) explore BPs in both classical and real-world formats but shift from an open-ended rule-formulation task to a classification-based approach. Youssef et al. (2022) approach BPs with a reinforcement learning setting for extracting meaningful representations and counterfactual explanations. However, despite these efforts, BPs remain an unsolved challenge.

**Benchmarks for VLMs.** Benchmarks specifically designed for VLMs usually involve complex tasks such as image captioning, scene or diagram understanding, visual question answering (VQA), or visual-commonsense reasoning (Masry et al., 2022; Yue et al., 2024; Rahmanzadehgervi et al., 2024; Kamath et al., 2023; Antol et al., 2015; Johnson et al., 2017; Hong et al., 2021; Zellers et al., 2019; Hudson & Manning, 2019; Qiao et al., 2024; Liu et al., 2024b). Yet, most of these only require simple reasoning abilities. More recent benchmarks have been introduced to probe advanced reasoning skills, *e.g.*, logical learning (Helff et al., 2025; Vedantam et al., 2021; Xiao et al., 2024), mathematical reasoning (Lu et al., 2024) or analogy-based visual reasoning (Chollet, 2019; Moskvichev et al., 2023; Zhang et al., 2019). Małkiński et al., 2025 considered BPs for evaluating VLMs, concentrating on open-ended and classification-based settings. Our work shares their open-ended focus but goes further by adding two additional tasks to examine specific abilities we hypothesize are vital for solving Bongard problems. While Małkiński et al., 2025 compare synthetic Bongard problems to a newly proposed real-world variant, we target the original Bongard set, pinpointing especially challenging cases and identifying concept-detection inconsistencies. Generally, the shift towards more cognitively demanding tasks is promising, comprehensive diagnostic evaluations of VLMs' reasoning capabilities that pinpoint sources of error and model limitations remain scarce. Furthermore, the degree to which these models genuinely comprehend complex, abstract visual concepts is yet to be fully investigated.

**Open-Ended Visual Reasoning.** Most existing benchmarks rely on classification tasks (Chollet, 2019; Helff et al., 2025; Vedantam et al., 2021), multiple-choice formats (Zhang et al., 2019; Lu et al., 2024), or synthetic environments with a limited set of operations (Moskvichev et al., 2023). While these setups tackle pattern recognition and visual reasoning, they do not fully capture real-world complexity. Open-ended visual puzzles, such as BPs, present a greater challenge by requiring models to map visual input to text without predefined rules or language biases. This lack of structured guidance makes the task more demanding, better reflecting real-world reasoning.

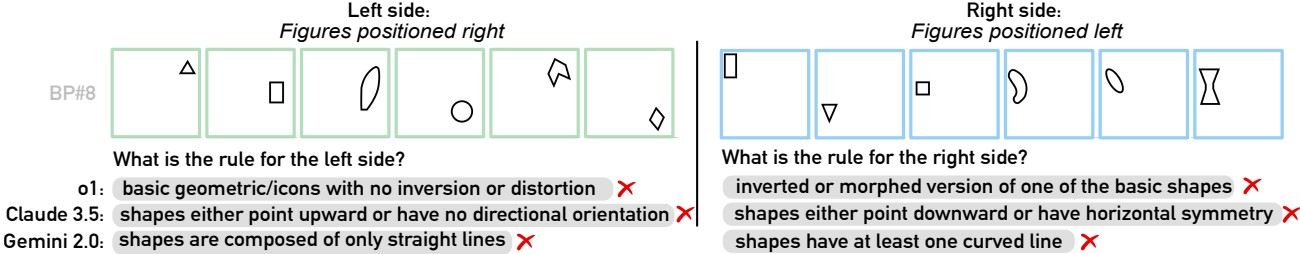

Figure 2: **VLMs struggle to solve BPs out of the box**. The images of BP#8 are depicted, where the left side has images with shapes positioned right and the right side has images of shapes positioned on the left. Although the concepts *left* and *right* may seem trivial to a human, the VLMs struggle to generate discriminative rules.

## 3. Bongard Problems and VLMs

**Bongard problems** (BP), introduced in 1970 by Mikhail Bongard (Bongard & Hawkins, 1970), are visual puzzles that test for capabilities of pattern recognition, concept formation, and abstraction. Each BP consists of twelve simple black-and-white diagrams divided into a left and a right group. While both sides may share similarities, each side has one distinct property or rule that is shared by all six images on that side but is not present in any image on the opposite side. An example BP is shown in Figure 2 where the distinguishing properties are *right* and *left* position.

The task is the linguistic expression of the underlying rule that distinguishes the two groups. These rules vary in complexity, ranging from simple geometric properties like the presence of a circle to more abstract or relational concepts like symmetry or the presence of a right angle. In some cases, the rule of the right side is just the negative of the left rule, like BP#24 (*a circle present* vs. *no circle present*). Still, for the majority of BPs the second rule is a more specific opposite of the first *e.g.*, BP#6 (*triangles* vs. *quadrangles*).

In contrast to mainstream classification tasks, BPs differ in their complexity and reliance on abstract reasoning rather than direct pattern recognition. Specifically, BPs test the ability to express distinctive and common features of images, including the pattern recognition necessary to correctly associate the features with images, as well as the ability to come up with textual rules that can characterize the meta-pattern (not within each but) across all twelve diagrams that constitute a BP.

This multi-modality of BPs makes them an interesting challenge for multimodal AI such as VLMs. However, the specific nature of these puzzles raises questions concerning the best experimental setup for VLM evaluation. In the following we therefore introduce several, different prompt strategies for VLMs to solve BPs.

**Task 1: Open-Ended Solving of Bongard Problems.** In this setting, for each BP, a model is prompted individually by providing a text prompt together with an image showing all twelve diagrams. The prompt follows the following setup. A model is given a text prompt alongside an image containing twelve black-and-white diagrams arranged into left and right sides. The prompt first describes the image structure, including the number and arrangement of diagrams. It then defines the task, explaining that each side follows a distinct rule that does not apply to the other. The model is instructed to analyze the diagrams step by step to infer these underlying rules. Finally, the required output format is specified as a dictionary with two entries, one for each rule. The expected response are two rules in natural language, one for the left side and one for the right side. The complete prompt can be found in Listing 1.

**Task 1.1: Multiple Choice Setting.** In this setting, rather than having the model generate the rules itself, we provide it with a set of predefined rule pairs from the BP domain. The model is then tasked with selecting the correct rule pair from these options. The prompt follows the same structure as in the previous setting but includes an additional list of available rules for the model to choose from. The expected answer is the ID corresponding to the correct rule-pair (cf. Listing 2).

**Task 2: Detect Specific Concepts.** To specifically investigate the limitations of visual descriptions we create an additional *perception* task. Here, the relevant concepts for the BP are provided as context (*e.g.*, *horizontal* and *vertical* orientation). Based on this, the task is to predict for every single image of the BP whether the left side or the right side concept is true.

To achieve this, we designed a tailored prompt for each BP that directly targets its ground truth concepts. For each of the 12 images in a BP, the prompt asks whether the concept from the left or right side of the problem applies to that specific image. To generate these prompts, we provide an LLM with example prompts for BP#16 and BP#55 (cf. Listing 4 and Listing 5) along with the corresponding ground truth rule[2]. Based on this input, the LLM automatically generates a custom prompt for each BP (cf. Listing 3).

---

[2] https://www.foundalis.com/res/bps/bongard_problems_solutions.htm

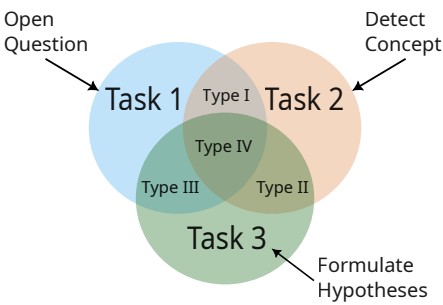

Figure 3: **Overview over our three different tasks and how model behaviour across them can intersect.** The subsets define different types of model behaviours.

**Task 3: Formulate Hypotheses.** To solve a BP, identifying the correct rule is essential. Even if a model can accurately evaluate whether a rule applies to a given BP, it cannot solve the BP unless it can also generate the rule itself. Thus, we propose a fourth evaluation setting in which the task is to hypothesize potential discriminative rules given a BP. Unlike the Open-Ended setting, these rules do not need to be true. Instead, this setting is more akin to a "brainstorming" exercise. The task is considered successful if the ground truth rule is included among the proposed hypotheses. A hypotheses should include a rule for both the left and right side of the BP and will only be valid, if both rules are correct. The prompt for this setting can be found in Listing 6.

**Investigating Different Model Behaviours.** With our different evaluation tasks we want to investigate the different behaviours of the models and how the successful solving of one tasks relates to the model being able to solve the other tasks. For this we differentiate between different subsets of the introduced tasks, Type I, Type II, Type III and Type IV in Figure 3, that result from the intersections of the solved BPs in the different tasks. These are defined in the following:

- Type I: A model is able to solve the BP and can classify images according to the underlying ground truth concepts correctly but fails to come up with the correct hypotheses in Task 3.

- Type II: A model can hypothesize the correct rule and is able to apply the rule correctly to all images. However, in the open-ended setting its not able to connect the dots to solve the problem out-of-the-box.

- Type III: A model is able to solve the BP and can formulate a correct hypothesis. However, it cannot classify the images of the BP correctly based on the ground truth rule.

- Type IV: A model is able to answer a BP correctly over all three task settings.

Ideally, one would expect, that if a model is able to solve Task 1, it would also succeed in the other tasks, leading to only Type IV subsets. If there occur subsets of the other types, this raises questions regarding the robustness and reasoning capabilities of the models. This will be evaluated in the following within the course of our evaluation.

## 4. Experimental Evaluation

Our experimental evaluations investigate to what extent state-of-the-art VLMs can solve Bongard problems. For this, we first evaluate the models quantitatively on all 100 puzzles based on the settings of Task 1 and Task 1.1. We then compare their performance against humans and investigate them qualitatively in more detail based on the settings of Task 2 and Task 3. We hereby address the following research questions:

**(Q1)** How well can state-of-the-art VLMs solve Bongard problems? (Task 1 + Task 1.1)

**(Q2)** How do VLMs perform in comparison to humans?

**(Q3)** How accurately can VLMs classify images of the BPs given the ground truth rule? (Task 2)

**(Q4)** How accurately can VLMs generate an underlying hypothesis? (Task 3)

**Data.** For our evaluations, we considered the 100 original Bongard problems of (Bongard & Hawkins, 1970). We used the dataset variation of (Depeweg et al., 2024), which contains high-resolution images of the original diagrams.

**Models.** We evaluated the proprietary models o1 (OpenAI, 2024b), GPT-4o (OpenAI, 2024a), Claude 3.5 Sonnet (Antropic, 2024), Gemini 2.0 Flash Experimental (Deepmind, 2025), Gemini 1.5 Pro (Team, 2024), and the open-source models Qwen2VL-72B-Instruct (Wang et al., 2024a), LLaVA-OneVision 72B (Li et al., 2025) and InternVL 2.5 78B (Chen et al., 2024). For simplicity, in the following the models are referred to as o1, GPT-4o, Claude, Gemini 2.0 and 1.5, Qwen2VL, LLaVA-OneVision, and InternVL 2.5 respectively. The specific models and their configurations are given in Suppl. A.2.

**LLM-as-a-Judge** To evaluate the open-ended responses of Task 1 and Task 3, we follow common practice and employ an LLM-as-a-Judge (Zheng et al., 2023) (from now on referred to as LLM-Judge) to automatically judge the responses. In our evaluations, we use the model GPT-4o for it. (*cf.* Suppl. A.3 for more details).

**Model Setup.** For all evaluations we tasked our selection of VLMs with solving each BP three times[3]. The answers in

---

[3]Exception: In Task 2 o1 was prompted once.

Table 1: **Performance of VLMs on 100 BPs (top) as well as multiple-choice BPs (bottom)**. Results depict the rounded average of solved BPs over 3 runs. All models struggled with the classical BP setup, with o1 achieving the highest score, solving only 43 out of 100 BPs. Even on the multiple-choice BPs, difficulties persist. Only when the number of choices is considerably limited does the performance increase.

| | o1 | GPT-4o | Claude 3.5 Sonnet | Gemini 2.0 Flash | Gemini 1.5 Pro | LLaVA-OneVision | Qwen2VL | InternVL 2.5 |
|---|---|---|---|---|---|---|---|---|
| **Solved BPs** (of 100) | **43** | 25 | 31 | 25 | 8 | 7 | 14 | 16 |
| Multiple Choice (100) | **57** | 23 | 37 | 27 | 16 | 6 | 11 | 21 |
| Multiple Choice (10) | **91** | 68 | 80 | 67 | 59 | 37 | 66 | 60 |

the context of Task 1 and Task 3 were evaluated by the LLM-judge, which determined whether each response correctly solved the BP. We provide the quantitative results of (Q1) as rounded averages and for the rest of the evaluations consider a BP solved if it is solved in two out of three attempts.

**Human Evaluation Setup.** For evaluating human performances on BPs, 30 participants were recruited among Prolific users (Prolific, 2014) to participate in an online experiment run via the online platform (LimeSurvey, 2012). The only selection criterion was English as a first language. The experiment was divided into three sessions, each taking approximately 30 minutes. In each session, participants were asked to solve 33 BPs. The BPs were shown in the original order (#2 to #100). BP#1 was used in the instructions. In each problem, participants had to provide a rule for each side of the BP or click "I don't know" and move on to the next BP. Participants were only invited to participate in the following session if they tried to answer more than 50 % of the BPs presented to them in the session. All participants were reimbursed for their time and could receive bonus payments for correct solutions to problems and completing all three study parts. 23 participants completed all the sessions. Out of these, we excluded three datasets: two because they did not try to answer more than 50 % of the BPs, and one for submitting nonsensical answers in the last session. The resulting dataset consists of the solution of 20 participants for 99 BPs each.

**Can VLMs solve Bongard problems? (Q1)** As a first step, we aim to assess how well current state-of-the-art VLMs can solve BPs according to Task 1, *i.e.*, the open-ended query setting. The evaluation results are shown in the top row of Table 1. Notably, across the set of VLMs, o1 emerged as the best-performing model, with an average of 43 solved BPs. This is followed by Claude with 31, and GPT-4o and Gemini 2.0 with 25 solved problems. This performance is still surprisingly low, especially when compared to human abilities (*cf.* (Q2) and (Linhares, 2000; Nie et al., 2020)). A more detailed breakdown of which BPs were solved is provided in Table 4 (Suppl.). Even comparatively simple BPs, such as *positioned right vs. left* (BP#8) and *thin and elongated vs. compact* (BP#11), remained unsolved in all attempts. Example responses from the VLMs for BP#8

(*cf.* Figure 2) reveal that the models fail to identify the correct rule and instead focus on irrelevant features, such as *shape distortion*, *horizontal symmetry*, and *curved lines*.

In an extended setup (Task 1.1), we evaluate how performance changes when the models are explicitly provided with all existing rule pairs of the BPs and asked to select the correct one (*cf.* Table 1 *Multiple Choice (100)*). Interestingly, this does not substantially change the results of GPT-4o, Gemini 2.0, LLaVA-OneVision, or Qwen2VL. However, o1, Claude, Gemini 1.5, and InternVL 2.5 do benefit from this setup, *e.g.*, Claude solves now 37 BPs on average, while o1 solves even 57.

To simplify the task, we reduce the number of answer choices to 10 possible rule pairs, ensuring that the true underlying rule is always included, and repeat the evaluation procedure using Task 1.1. In this simplified setting, the selected models show significant improvement, solving up to 91 BPs (*cf.* Table 1, *Multiple Choice (10)*). Notably, the correct solution was already present among the choices in the previous setting (*Multiple Choice (100)*), where the models previously selected incorrectly. This raises an important question: Do the models genuinely grasp the underlying concepts, or is the improvement merely a byproduct of a less complex elimination process? The results suggest that simplification helps models better leverage contextual cues, but does not conclusively demonstrate conceptual understanding. This leads to a broader inquiry: Can specialized contextual cues enable a model to solve a BP that it previously failed to solve? We explore this in our (Q3) evaluations. Before that, however, let us first compare the results of the VLMs to human reasoning abilities.

**How do VLMs perform in comparison to humans? (Q2)** We next evaluate how well the individual VLMs compare to human performance on the BPs. Hereby, for a more detailed comparison, we group the BPs into five categories based on the nature of their ground truth rules (*cf.*, Table 4 (Suppl.) for the categorization). The results of the human evaluations over all BPs are reported in Figure 4 (left), where we have added the corresponding VLM performances (over all models and runs) for comparison. We observe that the best human performance surpasses VLM performance by

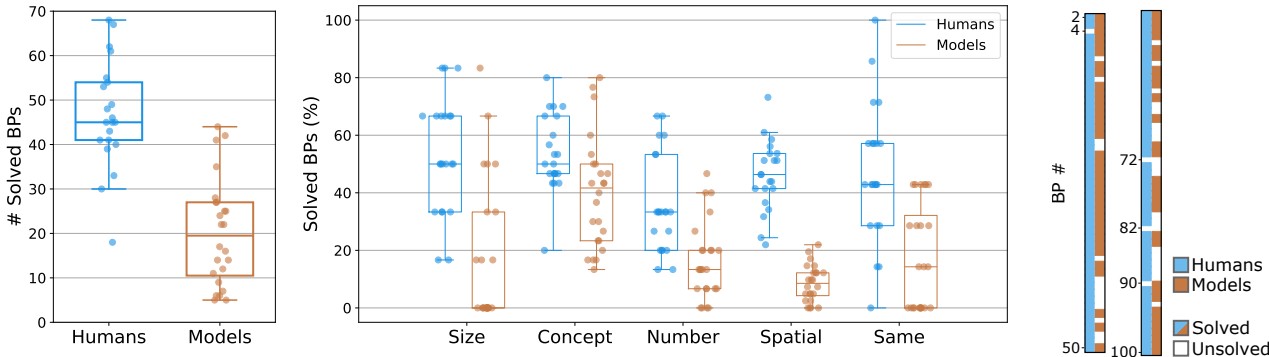

Figure 4: **Human results compared to results of VLMs.** (Left): Comparing the total number of solved BPs between human participants and VLMs. (Middle): Breakdown of solved BPs by category. (Right): Heatmap illustrating which BPs were solved by the set of human participants and VLMs overall. Humans and models tend to solve different problems.

far, *i.e.*, 68 BPs were solved. Interestingly, we see that the performance varies a lot between the subjects. However, there is a substantial number of them able to solve 50 or more of the problems.

When focusing on the different BP categories (*cf.* Figure 4 middle), we observe that humans are substantially better at solving BPs that have spatial reasoning components. *I.e.*, *spatial* is one of the trickiest categories for current VLM models, with all investigated models solving under 25%. For comparison, the top-performing humans solve more than 60%. This is in line with recent works suggesting that VLMs struggle with spatial relations (Rahmanzadehgervi et al., 2024; Kamath et al., 2023; Zhang et al., 2024).

In the *concept* category, o1 scored the highest accuracy with ∼ 77% solved of these BPs (*cf.* Figure 8 for the detailed results). This is a category where humans appeared to struggle more, *i.e.*, solved on average 54% of the BPs. This could be due to that VLMs hereby benefit from their extensive world knowledge while not all humans might be familiar with geometrical visual concepts, such as *convex* vs. *concave* (BP#4) and *thin elongated convex hull* vs. *compact convex hull* (BP#12).

As indicated by the distinct comparison across the BP categories, the BPs solved by the models differ from the ones solved by the humans. If we aggregate the values of the study participants and compare which BPs they solved (more than half of them found a solution) against which o1 solved (in at least 2 attempts) we see that there are 16 BPs that o1 solved but not the humans. On the other hand, the humans solved 31 BPs that o1 was not able to solve.

When comparing BPs that have been solved at least once by any VLM to those solved at least once by a human, an interesting pattern emerges, as visualized in Figure 4 (right). Out of the 100 BPs, nearly all are solved by at least one human participant. In contrast, only 66 BPs are solved at

least once by any model. A more detailed breakdown of the model's results can be found in Figure 8 (bottom). The results show that there exist a group of BPs that remains particularly challenging for current state-of-the-art VLMs.

In summary, we must conclude in response to (Q2) that even though VLMs exceed humans in some of the problems, they perform significantly worse in the overall picture, falling short of matching the visual pattern recognition and reasoning capabilities of humans. To better understand the shortcomings, we now conduct a detailed analysis to determine whether they can recognize the core concepts of the BPs when explicitly prompted for them in (Q3).

**Can VLMs detect concepts of BPs?** **(Q3)** In the previous evaluations, we observed that the investigated VLMs performed poorly on the BP dataset. This could be due to difficulties in accurately perceiving the diagrams, as well as reasoning failures, such as incorrectly formulating rules that apply differently to each side. To investigate this in more detail, we next evaluate the models according to Task 2. Recall that in this setting, a model is asked to classify the single images of a BP based on the ground truth concepts for the left and the right side.

Intuitively, one might expect that if a model can solve a BP in Task 1, it should also be able to solve Task 2 for this BP. To examine this, we compare the set of BPs solved in Task 1 to those successfully completed in Task 2. In Task 1, we define a BP as solved if it is correctly identified in at least two out of three attempts. Similarly, in Task 2, a BP is considered solved if all its images are correctly classified in at least two out of three attempts. We visualize these two sets and their intersection in Figure 5. Surprisingly, we find that the overlap between these two sets is quite small. This suggests that for a substantial subset of BPs, the models are able to solve them in Task 1 despite failing to classify every individual image correctly according to the ground

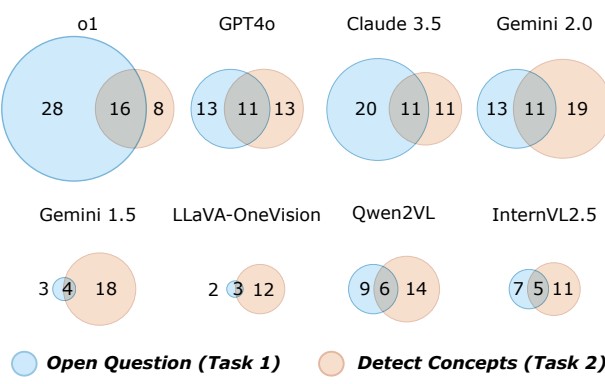

Figure 5: **Comparison of VLM performance on Task 1 and Task 2.** Blue circles represent the set of BPs solved in Task 1 (open-ended solving); orange circles those correctly classified in Task 2 (concept recognition). The overlapping region indicates BPs where a model succeeded in both.

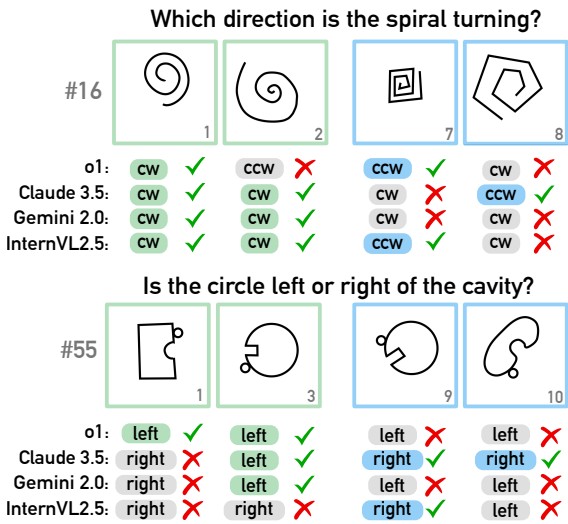

Figure 6: **VLMs fail to identify simple visual concepts (Task 2).** VLMs are challenged with identifying visual concepts in the individual images of the BPs. Abbreviations used for *clockwise (cw)* and *counter-clockwise (ccw)*.

truth rule. Conversely, there are also many BPs where the models perfectly classify images but still fail to solve the open-ended task. This large discrepancy highlights a surprising gap between recognizing correct classifications and effectively applying that knowledge in problem-solving. We report the number of correctly identified concepts for Task 2 in Table 9 (Suppl.) and further investigate in Suppl. B.4 whether the observed gap is affected by presenting full BP images versus individual images to the VLM.

For a better assessment of these results, we give a qualitative impression of the models' mistakes when prompted to identify specific, rather simple, concepts of BP#16 and BP#55 in Figure 6 (*cf.* Table 10, 11 for all responses and Figure 9 for additional BP#29 and #37). We find that for BP#16 (*cf.* Figure 6 top), even though some images are classified correctly, none of the models can classify all images correctly. Instead, we can see a tendency to classify one of the directions rather than the other, *e.g.*, Claude and GPT-4o almost always predict *clockwise* as the turning direction for all of the diagrams.

For BP#55 (*cf.* Figure 6 bottom) we also see that none of the models is able to classify the concepts of all images correctly. Here, we can see very similar behaviors across the different models, *e.g.*, almost all models categorize the first diagram of the BP falsely, but in most of the attempts, they are able to categorize the third diagram correctly. Interestingly, we found a strong correlation between the absolute position of the circle in the diagram (*left* vs. *right*) and the models responses. We take this as an indicator, that the models did not manage to reason spatially about the position of the circle in relation to the cavity.

When we consider the overall number of solved concepts across the models (*cf.* Table 9), we see that weaker models from Task 1 achieve slightly better results in this setting,

while, surprisingly, the stronger models perform relatively worse compared to their performance in Task 1. *E.g.*, Gemini 2.0 solves the most problems in this setting, with only 30 correct concepts. This unexpected trend suggests that higher-performing models in the open-ended task do not necessarily excel in applying ground truth concepts to individual images. Overall, the number of puzzles in which the models correctly identify the presence of the provided ground truth concept is substantially low, highlighting a persistent challenge in concept recognition and classification.

Conclusively, the observed behavior is remarkable and seems to indicate that perception is a key issue for not identifying the correct rules of BPs. With regard to (Q3) we have to conclude that the current perception capabilities of the evaluated VLMs are still insufficient to visually capture the concepts required for solving BPs.

**Can VLMs generate relevant hypotheses? (Q4)** In our next task evaluation, we investigate the potential of the models for generating correct rules for the BPs. In (Q3), we observed that the investigated models failed to classify the BP images accurately. This raises the question of whether the models might be finding the correct rule for most problems but mainly failing to apply it properly to individual BP images, ultimately overlooking it. To explore this, we evaluate the models on Task 3 and report the results in Figure 7 and Table 15.

Surprisingly, we observe that the set of correctly hypothesized rules in Task 3 is significantly larger than the number of solved BPs in Task 1. For instance, o1 proposes correct hypotheses in 72 out of 100 BPs while only solving 44 open

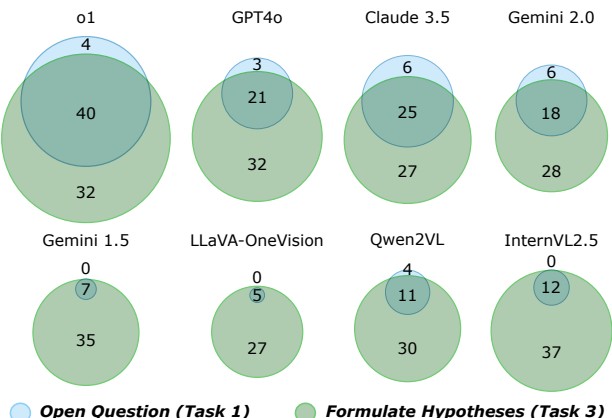

Figure 7: **Comparison of VLM performance on Task 1 and Task 3.** Blue circles represent the set of BPs solved in Task 1 (open-ended solving); green circles those correctly classified in Task 3 (formulate hypotheses). The overlapping region indicates BPs where a model succeeded in both.

question BPs. Such a disparity is particularly pronounced for VLMs that initially performed poorly in Task 1, such as LLaVA-OneVision which manages to generate 32 correct hypotheses while solving 5 open question BPs.

Furthermore, we observe that for most models, the set of BPs solved in Task 1 is not a subset of those with correctly identified rules in Task 3. This is somewhat unexpected, as one might assume that successfully solving a BP would imply the correct rule is among the hypotheses generated in Task 3. This finding may point to a lack of robustness in generalizing across the different task setups.

In conclusion, the results suggest that while VLMs may struggle to solve BPs directly, they more often succeed in formulating correct hypotheses about the underlying rules. However, this process might be brittle, as for some models solved BPs from Task 1 are not solved in Task 3. With regard to (Q4), we conclude that while hypothesizing correct rules seems to be easier for the models than solving the BP straight ahead, they are still not able to come up with correct hypotheses for all the problems.

## 5. Discussion: Remaining Challenges for AI

Based on the results of the previous evaluations which are summarized jointly in Figure 10, there are several additional important aspects to address. Specifically, we want to investigate the relations between the solved BPs in our proposed task settings (*cf.* Figure 3).

The first notable observation is that there are almost no Type I subsets, suggesting that if a model can solve the BP and correctly classify the concept, it is also likely to generate the correct ground truth rule in Task 3. However,

the presence of a significant number of Type II subsets is surprising. This implies that while the models can formulate a correct hypothesis and correctly classify BP images according to the ground truth rule, they still fail to solve the BP in the open-ended task. This suggests that, although the models theoretically possess the necessary knowledge, they struggle to apply it effectively.

When examining Type III subsets, we find that they make up a substantial portion of the diagrams, particularly for the stronger models. This is intriguing because it suggests that the models can determine the correct rules even in the open-ended task, yet they fail to apply it correctly to individual BP images (further undermining the findings of (Q2)). This raises the question of why this discrepancy occurs. One possible explanation is that the proposed rules are not derived through explicit reasoning. If the models were truly reasoning based on the rules, we would expect the individual images to be classified correctly. Instead, it seems that the models may infer the most likely rules based on the overall set of images without actually executing these rules for each image, leading to occasional success despite the underlying inconsistency.

Interestingly, we find that the proportion of Type IV BPs (*i.e.*, solving a BP across all three tasks) across the models is relatively low compared to the number of solved BPs in Task 1. For example, o1 has only 14 Type IV BPs out of 44 solved BPs, representing just around 32%. While this ratio is slightly better for the weaker models in Task 1, the overall number of solved BPs in these models remains low, making the comparison less meaningful (*cf.* Table 15).

**Most challenging BPs.** Throughout the experiments, we saw that the investigated VLMs can solve some BPs better than others, where the set of solved puzzles appears to differ across task setting. Interestingly, we find that there are 10 BPs that are especially hard, *i.e.*, they were not solved in any of the tasks (*cf.* Suppl. B.7). We think that these BPs can serve as a good foundation for further research and bench-marking.

**Limitations.** While BPs are valuable for assessing abstract reasoning, they also represent a narrow and highly specialized set of challenges that may not comprehensively reflect the broad range of problems VLMs encounter in real-world applications. Even though we take the less distributed dataset of the BPs (high resolution), it is still possible that the problems might be part of the training data. Future work should expand these evaluations to new and more diverse tasks. Additionally, the reliance on our LLM-Judge introduces some uncertainty in the evaluation process, which we discuss in Suppl. A.3.

## 6. Conclusion and Future Work

This work presented a diagnostic evaluation of VLMs using the classical Bongard problems (BPs), providing valuable insights into their current capabilities of pattern recognition and abstract reasoning. Our experimental results highlight a significant gap between human-like visual reasoning and machine cognition. Specifically, we found that VLMs are still largely unable to solve the majority of BPs, with the best-performing model, o1, solving only 43 out of the 100 BPs. Moreover, our analysis suggests that the limitations of current VLMs extend beyond just visual reasoning; they also struggle to perceive and comprehend elementary visual concepts, such as simple spirals.

Our findings raise several critical questions: Why do VLMs encounter difficulties with seemingly simple Bongard Problems, despite performing impressively across various established VLM benchmarks (Li et al., 2024; Duan et al., 2024)? How meaningful are these benchmarks in assessing true reasoning capabilities? Future work could use our results as base to develop new benchmarks. Further, the discovered Type II subsets suggest opportunities for improving the performance, *e.g.*, via a multi-stage reasoning approach.

## Acknowledgements

The authors would like to thank Frank Jäkel for useful discussions. This work was supported by the Priority Program (SPP) 2422 in the subproject "Optimization of active surface design of high-speed progressive tools using machine and deep learning algorithms" funded by the German Research Foundation (DFG). We acknowledge support of the hessian.AI Service Center (funded by the Federal Ministry of Education and Research, BMBF, grant No 01IS22091), the "Third Wave of AI", and "The Adaptive Mind". The work was also supported by the LOEWE research priority program "WhiteBox" [grant number LOEWE/ 2/13/519/03/06.001(0010)/77] (funded by the Hessian Ministry of Higher Education, Research, Science and the Arts). Further, this work was funded by the European Union (Grant Agreement no. 101120763 - TANGO). Views and opinions expressed are however those of the author(s) only and do not necessarily reflect those of the European Union or the European Health and Digital Executive Agency (HaDEA). Neither the European Union nor the granting authority can be held responsible for them. The authors of the Eindhoven University of Technology received support from their Department of Mathematics and Computer Science and the Eindhoven Artificial Intelligence Systems Institute.

## Impact Statement

This study critically examines the reasoning capabilities of VLMs in the context of Bongard problems, a benchmark for human-like pattern recognition and abstract reasoning. By highlighting the limitations of VLMs in detecting and understanding fundamental visual concepts, our work provides insights into the gaps that remain between human cognition and machine perception and reasoning. These findings contribute to the broader discussion on the reliability and interpretability of AI models, particularly in tasks requiring abstraction and generalization.

From an ethical standpoint, our research underscores the risks associated with overestimating the reasoning abilities of VLMs, which could lead to premature deployment in critical applications such as medical diagnosis, autonomous systems, and decision-making tasks. As VLMs continue to be integrated into real-world applications, it is essential to remain cautious about their limitations and to develop methods for improving their interpretability and robustness.

While our work does not present immediate societal risks, it highlights the importance of ongoing evaluation and responsible deployment of AI systems, ensuring that their capabilities are well understood before being applied in high-stakes environments.

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

## Supplementary Materials

In the following, we provide details on the experimental evaluations as well as additional results.

## A. Experimental Details

We provide here the prompts for our evaluations (Suppl. A.1), model details (Suppl. A.2), details about the LLM-Judge (Suppl. A.3) and details about our human study (Suppl. A.4).

### A.1. Prompts for the experiments

In the following, the prompts used during the experiments are provided. The prompt for open-ended solving (Task 1) is shown in Listing 1. The prompt for the multiple choice setting (Task 1.1) is in Listing 2. The prompts for Task 2 and Task 3 are provided in Listing 3 and Listing 6.

---

**Listing 1** Prompt for Task 1. The model is asked to provide rules for the left side and the right side images of the Bongard problem.

---

```
1 You are provided with a black-and-white image consisting of 12 simple diagrams. Each diagram represents shapes with
  specific features, such as geometric properties or higher-level concepts.
2
3 - The 6 diagrams on the left side belong to Set A.
4 - The 6 diagrams on the right side belong to Set B.
5
6 ## Task:
7
8 Your task is to determine two distinct rules:
9
10 1. Set A Rule: Identify a rule that applies to all diagrams in Set A.
11 2. Set B Rule: Identify a separate rule that applies to all diagrams in Set B.
12
13 Important: The rule for Set A must not apply to any diagram in Set B, and the rule for Set B must not apply to any
  diagram in Set A.
14
15 ## Step-by-Step Process:
16
17 1. Diagram Analysis: Carefully describe each diagram in detail, noting any geometric properties, patterns, or
  conceptual features.
18 2. Rule Derivation: Based on your analysis, deduce the rule for Set A and the rule for Set B, ensuring that each
  rule is unique to its set.
19
20 ## Final Answer Format:
21
22 Provide the final answer using the following format:
23
24 ```python
25 answer = {
26     'set A rule': '[LEFT RULE]',
27     'set B rule': '[RIGHT RULE]'
28 }
29 ```
30
31 Ensure that the rules are clearly defined, concise, and do not overlap between the sets.
```

---

### A.2. Model details

We provide details on the models used in our experiments in Table 2.

### A.3. LLM-Judge

For the systematic and automated evaluation of Task 1 and Task 3 we made use of LLM-as-a-Judge (Zheng et al., 2023). For this, we designed a prompt that explains the task in detail and gives five example judgements as directive to improve the performance via few-shot (*cf.* Listing 7 and Table 3 for examples). The answers to the reasoning task are compared to

**Listing 2** Prompt used in multiple choice experiment for solving BPs with solution options provided. The model is asked to select the rules for the left side and the ride side images of the BP that fits best. <SOLUTIONS> is replaced by a dictionary of the possible solutions the model can select from (either all 100 or a subset of 10).

```
1  You are provided with a black-and-white image consisting of 12 simple diagrams. Each diagram represents shapes with
   specific features, such as geometric properties or higher-level concepts.
2
3  - The 6 diagrams on the left side belong to Set A.
4  - The 6 diagrams on the right side belong to Set B.
5
6  Additionally, you are given a list of possible rule pairs, one of which is true for this image. Your goal is to
   identify the correct rule pair based on the features of the diagrams in Set A and Set B.
7
8  ## Task:
9
10 Your task is to identify and select the correct rule pair that is true for the sets. The rule pair is structured as
   follows:
11
12 1. Rule part 1: This rule should apply to all diagrams in Set A
13 2. Rule part 2: This rule should apply to all diagrams in Set B
14
15 Important: The rule for Set A must not apply to any diagram in Set B, and the rule for Set B must not apply to any
   diagram in Set A.
16
17 ## Step-by-Step Process:
18
19 1. Diagram Analysis: Carefully describe each diagram in detail, noting any geometric properties, patterns, or
   conceptual features.
20 2. Rule Derivation: Based on your analysis of the diagrams, select one rule from the provided list for Set A and a
   different rule for Set B.
21
22 ## Available Rules
23 You can choose from the following rule pairs:
24
25 <SOLUTIONS>
26
27 ## Final Answer Format:
28
29 Provide the final answer using the following format:
30
31 ```python
32
33 answer = {
34     'answer': <Solution ID>,
35 }
36 ```
37 Where <Solution ID> is the number corresponding to the correct rule pair that fits the criteria.
```

the ground truth BP rules[4]. To validate the automated results, we conduct manual annotations on a large subset, *i.e.*, 696 BP responses, both from the human study and the VLM answers. The human annotations were obtained from two experts that were recruited in-house. The two experts annotated the rules independently of the judge. We found that the judge has a 91% agreement score with the human judgement. One caveat in this method is that, in principle, many rules can be found for one Bongard problem (Depeweg et al., 2024), and each rule can be expressed in many ways that differ in context and specificity. We acknowledge that the LLM-Judge evaluates the answers only within the context of the ground truth rules and might misjudge technically correct rules. However, constraining the rule evaluation to the ground truth rules captures the pragmatics of Bongard's original rules.

### A.4. Human Study on BPs

Depeweg et al., 2024 evaluated in their work already humans solving BPs in the open-ended task. Their results showed a high human performance; the five subjects that worked on all BPs solved 42 of the 49 (the first problem excluded for comparison) on average (Min: 40, Max: 45). In comparison, the twenty participants in this study only solved 20 BPs out of the first 49 BPs correctly (Min: 7, Max: 31). The difference in performance can be the result of multiple factors. While the dataset from Depeweg et al., 2024 was recorded in a lab environment with university students, the participants in this study

---

[4]https://www.foundalis.com/res/bps/bongard_problems_solutions.htm

**Listing 3** Prompt for classifying ground truth concepts in single images (Task 2).

```
1 You are given a pair of contrary concepts. Your task is to craft a single prompt that asks which of those concepts
   appears in an image.
2
3 # Examples for reference:
4
5 ## Example 1:
6 Contrary Concepts: [
7     "Spiral curls counterclockwise",
8     "Spiral curls clockwise"
9 ]
10 Prompt: {spiral_promt}
11
12 ## Example 2:
13 Contrary Concepts: [
14     "A circle is at the left of the cavity if you look from inside the figure",
15     "A circle is at the right of the cavity if you look from inside the figure"
16 ],
17 Prompt: {cavity_prompt}
18
19 ## Your task:
20 Please formulate a new prompt for the contrary concepts below that asks which one is present in the given image.
21 Contrary Concepts: {value}
```

**Listing 4** Prompt for concepts of BP#16.

```
1 Your task is to determine the direction in which a spiral depicted in a 2D black and white diagram is turning.
2 The given diagram shows a spiral-like shape. In which direction is the spiral turning, starting from the center?
3
4 Please decide carefully whether the spiral is turning in clockwise or counterclockwise direction. Take a deep breath
   and think step-by-step. Give your answer in the following format:
5 ```
6 answer = {
7     "direction": <your answer>
8 }
9 ```
10 where <your answer> can be either "counterclockwise" or "clockwise".
```

were sampled from the prolific participant pool, which offers a broad range of educational and experience backgrounds, which could explain the higher variability (Paolacci & Chandler, 2014). Additionally, participants in our study might have optimized their time over their performance. On average, participants spent 49 seconds with a problem before skipping it. The experimental design of Depeweg et al., 2024 allowed skipping only after 2 minutes, forcing the participants to spend more time on problems.

**Listing 5** Prompt for concepts of BP#55.

```
1 Your task is to determine the position of a circle in relation to a cavity in a 2D black and white diagram.
2
3 The given diagram shows a shape with a cavity. From inside the figure, you need to decide if the circle is on the
  left or the right of the cavity. Carefully analyze the diagram step-by-step to identify the correct side.
4
5 Please decide carefully. Take a deep breath and think step-by-step. Give your answer in the following format:
6
7 ```python
8 answer = {
9     "position": <your answer>
10 }
11 ```
12 where <your answer> can be either "left" if the circle is to the left of the cavity or "right" if the circle is to
  the right of the cavity.
```

**Listing 6** Prompt for hypotheses (Task 3).

```
1 You are provided with a black-and-white image consisting of 12 simple diagrams. Each diagram represents shapes with
  specific features, such as geometric properties or higher-level concepts.
2
3 - The 6 diagrams on the left side belong to Set A.
4 - The 6 diagrams on the right side belong to Set B.
5
6 ## Task
7
8 Propose multiple distinct hypotheses explaining how the diagrams in Set A might share a common rule (or set of
  rules) that does not apply to Set B, and vice versa. In other words, each hypothesis should outline:
9
10 1. A feature or rule that unifies all 6 diagrams in Set A
11 2. A different feature or rule that unifies all 6 diagrams in Set B
12
13 These rules must be mutually exclusive - whatever defines Set A must not be true for Set B, and vice versa. You may
  consider a variety of factors, like shape type, number or count of shapes, orientation, symmetry, color usage, line
  style, or any other relevant factor - but only in the context of the 6 images of each set.
14
15 ## Answer Format
16
17 1. Return your hypotheses as a Python list of pairs of strings.
18 2. Each list element should be a tuple `(rule_for_Set_A, rule_for_Set_B)`.
19 3. For example:
20 ```python
21     [
22         ("All the diagrams contain at least one triangular shape", "There exists no triangular shape in the
          diagrams"),
23         ("Each diagram features a single shape that is positioned in the bottom half of the diagram", "Each diagram
          features a single shape that is positioned in the upper half of the diagram"),
24         ...
25     ]
26 ```
27 4. Ensure each rule is clear, specific and unique.
28 5. Avoid stating which hypothesis seems most likely or correct; simply propose a variety of possibilities.
29
30 ## Additional Guidelines
31
32 - Each hypothesis should stand on its own and be mutually exclusive to the other set.
33 - Be creative, but only base your statements on plausible observations from the 12 diagrams.
34 - Provide 20 distinct, well-defined hypotheses.
```

Table 2: Details on the models used in the evaluations.

| Model | Version | Parameters | Framework | Devices |
|---|---|---|---|---|
| o1 | o1-2024-12-17 | Default, max_tokens=2048 | openai (API) | - |
| GPT-4o | gpt-4o-2024-08-06 | Default, max_tokens=2048 | openai (API) | - |
| Claude | claude-3-5-sonnet-20241022 | Default, max_tokens=2048 | anthropic (API) | - |
| Gemini 2.0 | gemini-2.0-flash-exp | Default, max_tokens=2048 | google (API) | - |
| Gemini 1.5 | gemini-1.5-pro | Default, max_tokens=2048 | google (API) | - |
| LLaVA-OneVision | models–lmms-lab–llava-onevision-qwen2-72b-ov-chat | Default, max_tokens=2048 | transformers | 3 GPUs (NVIDIA A100-SXM4-80GB) |
| Qwen2VL | models–Qwen–Qwen2-VL-72B-Instruct | Default, max_tokens=2048 | transformers | 3 GPUs (NVIDIA A100-SXM4-80GB) |
| InternVL2.5 | models–OpenGVLab–InternVL2_5-78B | Default, max_tokens=2048 | llm_deploy | 4 GPUs (NVIDIA A100-SXM4-80GB) |
| GPT-4o (LLM-Judge) | gpt-4o-2024-08-06 | Default, temperature=0.0 max_tokens=2048 | openai (API) | - |

**Listing 7** Prompt for LLM-judge used across the experiments. There are five example judgements provided (<EXAMPLES> is placeholder). The judge needs to decide whether the answer is correct based on the provided ground truth (1) or incorrect (0).

```
1  You will be given a correct answer that states a rule for the left side and a rule for the right side of a visual
   pattern or scenario. You will also be given an answer from a model that attempts to describe these rules. Your task
   is to evaluate whether the model's answer accurately reflects the intent and essence of the correct answer.
2  # Evaluation Criteria:
3
4  1. Semantic Accuracy: Does the model's answer convey the same underlying concept or rule as the correct answer, even
   if the wording differs?
5  2. Logical Consistency: Is the model's answer logically consistent with the correct answer's rules?
6  3. Relevance: Does the model's answer directly address the rules provided in the correct answer?
7
8  # Response Instructions:
9
10 - Respond with "answer": 1 if the model's answer is correct according to the criteria above.
11 - Respond with "answer": 0 if the model's answer is incorrect.
12 - If the model's answer is only partially correct, consider whether the partial match sufficiently conveys the
   intended rule. If it does, respond with "answer": 1; otherwise, respond with "answer": 0.
13
14 # Examples:
15 <EXAMPLES>
16
17 Use the format above to judge the correctness of the model's answer based on the given correct answer.
18
19 # Task
20 - Correct Answer:
21     - Left: LEFT_RULE_SOLUTION
22     - Right: RIGHT_RULE_SOLUTION
23 - Model Answer:
24     - Left: LEFT_RULE_ANSWER
25     - Right: RIGHT_RULE_ANSWER
26 - Response:
27
```

Table 3: Examples used as few-shot examples for the LLM-Judge.

| | Correct Answer Left | Correct Answer Right | Model Answer Left | Model Answer Right | Score |
|---|---|---|---|---|---|
| Example 1 | Round outlined shapes | Filled squares | White circles | Black squares | 1 |
| Example 2 | Large shapes | Small shapes | Circular shapes | Irregular shapes | 0 |
| Example 3 | Square on top of circle | Circle on top of square | square top | square bottom | 1 |
| Example 4 | Circle | Triangle | Circle | Non-circle | 0 |
| Example 5 | Smooth contour figures | Twisting contour figures | smooth | jagged or zig-zag | 1 |

Table 4: Results of each VLM on the individual Bongard problems compared to the results of our human study. Each model was prompted three times and the number of correct responses is reported (of 3). The reported ratios are marked based on four intervals: less than 1/3 (white), [1/3-2/3) (light green), [2/3-3/3) (green), 3/3 (dark green).

| BP# | Categories | o1 | GPT-4o | Claude 3.5 | Gemini 2.0 | Gemini 1.5 | LLaVA-OV | Qwen2VL | InternVL 2.5 | Human |
|---|---|---|---|---|---|---|---|---|---|---|
| 1 | concept | 3/3 | 2/3 | 3/3 | 3/3 | 3/3 | 3/3 | 3/3 | 3/3 | - |
| 2 | size | 3/3 | 3/3 | 3/3 | 1/3 | 0/3 | 1/3 | 0/3 | 0/3 | 16/20 |
| 3 | concept | 3/3 | 3/3 | 3/3 | 3/3 | 3/3 | 2/3 | 3/3 | 2/3 | 19/20 |
| 4 | concept | 2/3 | 0/3 | 0/3 | 0/3 | 0/3 | 2/3 | 0/3 | 2/3 | 0/20 |
| 5 | concept | 3/3 | 2/3 | 3/3 | 3/3 | 3/3 | 3/3 | 3/3 | 3/3 | 16/20 |
| 6 | concept | 2/3 | 3/3 | 1/3 | 3/3 | 0/3 | 2/3 | 3/3 | 2/3 | 15/20 |
| 7 | concept | 3/3 | 1/3 | 3/3 | 3/3 | 0/3 | 0/3 | 0/3 | 1/3 | 17/20 |
| 8 | spatial | 0/3 | 0/3 | 0/3 | 0/3 | 0/3 | 0/3 | 0/3 | 0/3 | 18/20 |
| 9 | concept | 0/3 | 0/3 | 2/3 | 0/3 | 0/3 | 1/3 | 0/3 | 0/3 | 15/20 |
| 10 | concept | 3/3 | 2/3 | 2/3 | 3/3 | 0/3 | 0/3 | 0/3 | 0/3 | 11/20 |
| 11 | concept | 0/3 | 0/3 | 0/3 | 0/3 | 0/3 | 0/3 | 0/3 | 0/3 | 7/20 |
| 12 | concept | 3/3 | 0/3 | 0/3 | 0/3 | 0/3 | 0/3 | 0/3 | 0/3 | 3/20 |
| 13 | concept | 3/3 | 1/3 | 0/3 | 2/3 | 2/3 | 0/3 | 0/3 | 0/3 | 12/20 |
| 14 | size | 3/3 | 0/3 | 0/3 | 0/3 | 0/3 | 0/3 | 0/3 | 0/3 | 10/20 |
| 15 | concept | 3/3 | 3/3 | 1/3 | 2/3 | 0/3 | 1/3 | 0/3 | 0/3 | 19/20 |
| 16 | spatial | 0/3 | 1/3 | 0/3 | 0/3 | 0/3 | 0/3 | 0/3 | 0/3 | 5/20 |
| 17 | concept | 2/3 | 0/3 | 2/3 | 0/3 | 0/3 | 0/3 | 0/3 | 1/3 | 2/20 |
| 18 | concept | 3/3 | 0/3 | 2/3 | 0/3 | 0/3 | 0/3 | 0/3 | 0/3 | 7/20 |
| 19 | concept | 2/3 | 0/3 | 0/3 | 0/3 | 0/3 | 0/3 | 0/3 | 0/3 | 6/20 |
| 20 | spatial | 0/3 | 0/3 | 0/3 | 0/3 | 0/3 | 0/3 | 0/3 | 0/3 | 6/20 |
| 21 | size | 0/3 | 0/3 | 0/3 | 0/3 | 0/3 | 0/3 | 0/3 | 0/3 | 8/20 |
| 22 | size | 1/3 | 0/3 | 1/3 | 0/3 | 0/3 | 0/3 | 0/3 | 0/3 | 2/20 |
| 23 | number | 1/3 | 3/3 | 3/3 | 2/3 | 2/3 | 0/3 | 0/3 | 0/3 | 18/20 |
| 24 | concept | 2/3 | 0/3 | 3/3 | 2/3 | 0/3 | 0/3 | 0/3 | 0/3 | 12/20 |
| 25 | concept | 1/3 | 3/3 | 1/3 | 1/3 | 0/3 | 0/3 | 0/3 | 1/3 | 19/20 |
| 26 | concept | 1/3 | 0/3 | 0/3 | 0/3 | 0/3 | 0/3 | 0/3 | 0/3 | 5/20 |
| 27 | number | 0/3 | 1/3 | 0/3 | 0/3 | 0/3 | 1/3 | 0/3 | 0/3 | 6/20 |
| 28 | number | 2/3 | 0/3 | 1/3 | 0/3 | 0/3 | 0/3 | 0/3 | 0/3 | 3/20 |
| 29 | number | 1/3 | 0/3 | 0/3 | 1/3 | 0/3 | 0/3 | 0/3 | 0/3 | 8/20 |
| 30 | concept | 3/3 | 3/3 | 3/3 | 0/3 | 1/3 | 0/3 | 0/3 | 2/3 | 14/20 |
| 31 | number | 2/3 | 0/3 | 2/3 | 1/3 | 0/3 | 0/3 | 0/3 | 2/3 | 11/20 |
| 32 | concept | 3/3 | 3/3 | 3/3 | 2/3 | 0/3 | 1/3 | 3/3 | 2/3 | 7/20 |
| 33 | concept | 3/3 | 3/3 | 2/3 | 0/3 | 0/3 | 1/3 | 0/3 | 0/3 | 2/20 |
| 34 | size | 2/3 | 0/3 | 0/3 | 0/3 | 0/3 | 0/3 | 0/3 | 0/3 | 14/20 |
| 35 | spatial | 1/3 | 0/3 | 0/3 | 0/3 | 0/3 | 0/3 | 0/3 | 0/3 | 10/20 |
| 36 | spatial | 2/3 | 0/3 | 3/3 | 1/3 | 0/3 | 0/3 | 0/3 | 1/3 | 16/20 |
| 37 | spatial | 0/3 | 0/3 | 0/3 | 0/3 | 0/3 | 0/3 | 0/3 | 0/3 | 4/20 |
| 38 | size | 2/3 | 0/3 | 2/3 | 0/3 | 0/3 | 0/3 | 0/3 | 0/3 | 14/20 |
| 39 | spatial | 0/3 | 0/3 | 1/3 | 2/3 | 0/3 | 0/3 | 0/3 | 1/3 | 14/20 |
| 40 | spatial | 0/3 | 0/3 | 0/3 | 0/3 | 0/3 | 0/3 | 0/3 | 0/3 | 8/20 |
| 41 | spatial | 0/3 | 0/3 | 0/3 | 0/3 | 0/3 | 0/3 | 0/3 | 0/3 | 9/20 |
| 42 | spatial | 0/3 | 0/3 | 0/3 | 0/3 | 0/3 | 0/3 | 0/3 | 0/3 | 13/20 |
| 43 | concept | 0/3 | 0/3 | 0/3 | 0/3 | 0/3 | 0/3 | 0/3 | 0/3 | 13/20 |
| 44 | spatial | 0/3 | 0/3 | 0/3 | 0/3 | 0/3 | 0/3 | 0/3 | 0/3 | 2/20 |
| 45 | spatial | 1/3 | 0/3 | 0/3 | 1/3 | 0/3 | 0/3 | 0/3 | 0/3 | 15/20 |
| 46 | spatial | 0/3 | 0/3 | 0/3 | 0/3 | 0/3 | 0/3 | 0/3 | 0/3 | 13/20 |
| 47 | spatial | 3/3 | 3/3 | 0/3 | 3/3 | 0/3 | 0/3 | 0/3 | 2/3 | 17/20 |
| 48 | spatial | 0/3 | 0/3 | 0/3 | 0/3 | 0/3 | 0/3 | 0/3 | 0/3 | 10/20 |
| 49 | spatial | 0/3 | 0/3 | 0/3 | 0/3 | 0/3 | 0/3 | 0/3 | 0/3 | 10/20 |
| 50 | concept | 3/3 | 0/3 | 3/3 | 0/3 | 0/3 | 0/3 | 0/3 | 0/3 | 9/20 |
| 51 | spatial | 0/3 | 0/3 | 0/3 | 1/3 | 0/3 | 0/3 | 0/3 | 1/3 | 7/20 |
| 52 | spatial | 1/3 | 0/3 | 0/3 | 0/3 | 0/3 | 0/3 | 0/3 | 0/3 | 14/20 |
| 53 | number | 3/3 | 0/3 | 0/3 | 1/3 | 0/3 | 0/3 | 3/3 | 1/3 | 10/20 |
| 54 | spatial | 0/3 | 0/3 | 1/3 | 0/3 | 0/3 | 0/3 | 0/3 | 0/3 | 1/20 |
| 55 | spatial | 0/3 | 0/3 | 0/3 | 0/3 | 0/3 | 0/3 | 0/3 | 0/3 | 9/20 |
| 56 | same | 0/3 | 0/3 | 2/3 | 0/3 | 0/3 | 0/3 | 0/3 | 0/3 | 6/20 |
| 57 | same | 2/3 | 1/3 | 1/3 | 2/3 | 0/3 | 0/3 | 0/3 | 1/3 | 13/20 |
| 58 | same | 0/3 | 0/3 | 0/3 | 0/3 | 0/3 | 0/3 | 0/3 | 0/3 | 2/20 |
| 59 | same | 3/3 | 3/3 | 2/3 | 2/3 | 0/3 | 1/3 | 0/3 | 0/3 | 17/20 |
| 60 | same | 1/3 | 0/3 | 1/3 | 2/3 | 0/3 | 0/3 | 0/3 | 0/3 | 9/20 |
| 61 | spatial | 1/3 | 0/3 | 1/3 | 0/3 | 0/3 | 0/3 | 3/3 | 0/3 | 13/20 |
| 62 | spatial | 0/3 | 0/3 | 0/3 | 0/3 | 0/3 | 0/3 | 0/3 | 0/3 | 4/20 |
| 63 | spatial | 0/3 | 0/3 | 0/3 | 2/3 | 0/3 | 0/3 | 0/3 | 0/3 | 15/20 |
| 64 | spatial | 0/3 | 0/3 | 0/3 | 0/3 | 0/3 | 0/3 | 0/3 | 0/3 | 2/20 |
| 65 | spatial | 0/3 | 0/3 | 0/3 | 0/3 | 0/3 | 0/3 | 0/3 | 0/3 | 12/20 |
| 66 | spatial | 2/3 | 1/3 | 0/3 | 1/3 | 0/3 | 0/3 | 0/3 | 0/3 | 9/20 |
| 67 | spatial | 0/3 | 0/3 | 0/3 | 0/3 | 0/3 | 0/3 | 0/3 | 0/3 | 4/20 |
| 68 | spatial | 0/3 | 0/3 | 0/3 | 0/3 | 0/3 | 0/3 | 0/3 | 0/3 | 1/20 |
| 69 | spatial | 0/3 | 0/3 | 0/3 | 0/3 | 0/3 | 0/3 | 0/3 | 0/3 | 14/20 |
| 70 | number | 2/3 | 0/3 | 0/3 | 0/3 | 0/3 | 0/3 | 3/3 | 0/3 | 9/20 |
| 71 | number | 3/3 | 0/3 | 3/3 | 1/3 | 0/3 | 0/3 | 0/3 | 1/3 | 13/20 |
| 72 | spatial | 0/3 | 0/3 | 0/3 | 0/3 | 0/3 | 0/3 | 0/3 | 0/3 | 0/20 |
| 73 | spatial | 0/3 | 0/3 | 0/3 | 0/3 | 0/3 | 0/3 | 0/3 | 0/3 | 2/20 |
| 74 | spatial | 0/3 | 0/3 | 0/3 | 0/3 | 0/3 | 0/3 | 0/3 | 0/3 | 2/20 |
| 75 | spatial | 0/3 | 0/3 | 0/3 | 2/3 | 0/3 | 0/3 | 0/3 | 0/3 | 15/20 |
| 76 | concept | 0/3 | 0/3 | 1/3 | 2/3 | 0/3 | 0/3 | 0/3 | 0/3 | 9/20 |
| 77 | same | 0/3 | 3/3 | 0/3 | 0/3 | 0/3 | 0/3 | 0/3 | 1/3 | 13/20 |
| 78 | spatial | 0/3 | 0/3 | 0/3 | 0/3 | 0/3 | 0/3 | 0/3 | 1/3 | 9/20 |
| 79 | spatial | 0/3 | 0/3 | 1/3 | 0/3 | 0/3 | 0/3 | 0/3 | 0/3 | 1/20 |
| 80 | same | 0/3 | 0/3 | 0/3 | 0/3 | 0/3 | 0/3 | 0/3 | 0/3 | 6/20 |
| 81 | spatial | 0/3 | 0/3 | 0/3 | 0/3 | 0/3 | 0/3 | 0/3 | 0/3 | 1/20 |
| 82 | concept | 0/3 | 0/3 | 0/3 | 0/3 | 0/3 | 0/3 | 0/3 | 0/3 | 0/20 |
| 83 | spatial | 3/3 | 3/3 | 3/3 | 0/3 | 0/3 | 0/3 | 3/3 | 1/3 | 10/20 |
| 84 | spatial | 3/3 | 3/3 | 3/3 | 2/3 | 0/3 | 0/3 | 1/3 | 1/3 | 20/20 |
| 85 | number | 0/3 | 0/3 | 0/3 | 0/3 | 0/3 | 0/3 | 0/3 | 0/3 | 8/20 |
| 86 | number | 0/3 | 0/3 | 0/3 | 0/3 | 0/3 | 0/3 | 0/3 | 0/3 | 3/20 |
| 87 | number | 0/3 | 0/3 | 0/3 | 0/3 | 0/3 | 0/3 | 0/3 | 0/3 | 4/20 |
| 88 | number | 0/3 | 0/3 | 0/3 | 0/3 | 0/3 | 0/3 | 0/3 | 0/3 | 7/20 |
| 89 | number | 0/3 | 0/3 | 0/3 | 0/3 | 0/3 | 0/3 | 0/3 | 0/3 | 1/20 |
| 90 | number | 0/3 | 0/3 | 0/3 | 0/3 | 0/3 | 0/3 | 0/3 | 0/3 | 0/20 |
| 91 | number | 3/3 | 0/3 | 0/3 | 0/3 | 0/3 | 0/3 | 0/3 | 0/3 | 9/20 |
| 92 | concept | 0/3 | 0/3 | 2/3 | 0/3 | 0/3 | 0/3 | 0/3 | 0/3 | 15/20 |
| 93 | spatial | 0/3 | 0/3 | 0/3 | 0/3 | 0/3 | 0/3 | 0/3 | 0/3 | 6/20 |
| 94 | spatial | 3/3 | 3/3 | 2/3 | 0/3 | 0/3 | 1/3 | 0/3 | 1/3 | 17/20 |
| 95 | concept | 3/3 | 3/3 | 0/3 | 0/3 | 0/3 | 0/3 | 3/3 | 3/3 | 17/20 |
| 96 | concept | 3/3 | 3/3 | 0/3 | 2/3 | 0/3 | 0/3 | 0/3 | 0/3 | 14/20 |
| 97 | concept | 3/3 | 3/3 | 3/3 | 3/3 | 0/3 | 0/3 | 2/3 | 3/3 | 17/20 |
| 98 | concept | 3/3 | 3/3 | 2/3 | 3/3 | 3/3 | 0/3 | 0/3 | 0/3 | 16/20 |
| 99 | spatial | 1/3 | 0/3 | 3/3 | 0/3 | 0/3 | 0/3 | 0/3 | 0/3 | 16/20 |
| 100 | concept | 3/3 | 3/3 | 3/3 | 1/3 | 0/3 | 1/3 | 0/3 | 1/3 | 6/20 |

# B. Additional Results

In the following the detailed results of the evaluations are presented.

### B.1. Task 1 and 1.1

In Table 4 the results of of each model on Task 1 are reported. Further, it includes the results of our human study. The detailed results for Task 1.1 are given in Table 5 (100 Options) and Table 6 (10 Options).

### B.2. Comparison VLMs vs. Humans

In Table 4 we compare the results of the VLMs on a subset of BPs against human results from our study. Each BP has been categorized based on the ground truth rules, the possible categories are size, concept, number, spatial and same. Size means that the rule is based on the size of one or multiple shapes in the BP. Concept means, that the BP tests for a specific, more abstract concept, e.g., *convex* and *concave*. Under number BPs are grouped that require some form of counting, e.g., *one* vs *two* shapes. Spatial takes into account BPs that require spatial reasoning, e.g., some *shape is on top of the other*. And finally same considers BPs that test for same-different reasoning, *e.g.*, *solid quadrangles are identical* vs. *different*.

In Figure 8, we report detailed results of our evaluations regarding (Q2). We provide the results of each model type individually over each three runs and also the top 5 human study participants individually. Corresponding numerical results are provided in Table 7.

### B.3. Task 2

For Task 2 we report the classification accuracy of each model for the ground truth concepts of each BP in Table 8. We provide quantitative results for Task 2 visually in Table 9 and discuss additional qualitative results for the evaluations of (Q3), *i.e.*, also for BP#29 and #37. For BP#29, the models were asked to determine whether there are more shapes inside or

Table 5: Results of each VLM on the individual Bongard Problems when provided with all possible solutions *(Multiple Choice (100))*. Each model was prompted three times and the number of correct responses is reported (of 3). The reported ratios are marked based on four intervals: less than 1/3 (white), [1/3-2/3) (light green), [2/3-3/3) (green), 3/3 (dark green).

| BP# | o1 | GPT-4o | Claude 3.5 | Gemini 2.0 | Gemini 1.5 | LLaVA-OV | Qwen2VL | InternVL 2.5 |
|---|---|---|---|---|---|---|---|---|
| 1 | 3/3 | 3/3 | 3/3 | 3/3 | 3/3 | 0/3 | 3/3 | 3/3 |
| 2 | 3/3 | 1/3 | 3/3 | 1/3 | 0/3 | 0/3 | 0/3 | 0/3 |
| 3 | 3/3 | 3/3 | 3/3 | 3/3 | 3/3 | 0/3 | 3/3 | 3/3 |
| 4 | 3/3 | 0/3 | 0/3 | 0/3 | 2/3 | 0/3 | 0/3 | 1/3 |
| 5 | 3/3 | 3/3 | 3/3 | 3/3 | 3/3 | 1/3 | 3/3 | 3/3 |
| 6 | 3/3 | 3/3 | 3/3 | 3/3 | 3/3 | 3/3 | 3/3 | 3/3 |
| 7 | 3/3 | 3/3 | 3/3 | 3/3 | 0/3 | 0/3 | 3/3 | 1/3 |
| 8 | 0/3 | 0/3 | 0/3 | 0/3 | 0/3 | 0/3 | 0/3 | 0/3 |
| 9 | 3/3 | 3/3 | 3/3 | 2/3 | 3/3 | 1/3 | 0/3 | 1/3 |
| 10 | 3/3 | 3/3 | 3/3 | 3/3 | 3/3 | 0/3 | 3/3 | 3/3 |
| 11 | 3/3 | 0/3 | 2/3 | 1/3 | 0/3 | 0/3 | 0/3 | 0/3 |
| 12 | 0/3 | 0/3 | 0/3 | 0/3 | 0/3 | 0/3 | 0/3 | 0/3 |
| 13 | 3/3 | 0/3 | 3/3 | 3/3 | 3/3 | 1/3 | 0/3 | 1/3 |
| 14 | 1/3 | 0/3 | 0/3 | 0/3 | 0/3 | 0/3 | 0/3 | 0/3 |
| 15 | 2/3 | 0/3 | 0/3 | 0/3 | 0/3 | 0/3 | 0/3 | 0/3 |
| 16 | 3/3 | 2/3 | 1/3 | 0/3 | 0/3 | 0/3 | 0/3 | 0/3 |
| 17 | 1/3 | 0/3 | 0/3 | 0/3 | 0/3 | 0/3 | 0/3 | 0/3 |
| 18 | 3/3 | 0/3 | 3/3 | 0/3 | 0/3 | 0/3 | 0/3 | 0/3 |
| 19 | 3/3 | 0/3 | 0/3 | 0/3 | 0/3 | 0/3 | 0/3 | 0/3 |
| 20 | 0/3 | 0/3 | 0/3 | 0/3 | 1/3 | 0/3 | 0/3 | 1/3 |
| 21 | 1/3 | 0/3 | 0/3 | 2/3 | 0/3 | 2/3 | 0/3 | 1/3 |
| 22 | 0/3 | 0/3 | 0/3 | 0/3 | 0/3 | 0/3 | 0/3 | 0/3 |
| 23 | 3/3 | 3/3 | 3/3 | 2/3 | 0/3 | 1/3 | 3/3 | 2/3 |
| 24 | 3/3 | 1/3 | 3/3 | 3/3 | 3/3 | 0/3 | 0/3 | 3/3 |
| 25 | 3/3 | 2/3 | 0/3 | 0/3 | 1/3 | 0/3 | 0/3 | 2/3 |
| 26 | 2/3 | 0/3 | 1/3 | 0/3 | 2/3 | 0/3 | 0/3 | 1/3 |
| 27 | 2/3 | 1/3 | 1/3 | 2/3 | 0/3 | 0/3 | 0/3 | 0/3 |
| 28 | 0/3 | 0/3 | 0/3 | 0/3 | 0/3 | 0/3 | 0/3 | 0/3 |
| 29 | 2/3 | 1/3 | 2/3 | 3/3 | 3/3 | 0/3 | 3/3 | 1/3 |
| 30 | 3/3 | 3/3 | 3/3 | 0/3 | 3/3 | 0/3 | 0/3 | 0/3 |
| 31 | 2/3 | 0/3 | 1/3 | 0/3 | 0/3 | 0/3 | 0/3 | 0/3 |
| 32 | 2/3 | 0/3 | 0/3 | 0/3 | 0/3 | 0/3 | 0/3 | 0/3 |
| 33 | 2/3 | 0/3 | 2/3 | 0/3 | 0/3 | 0/3 | 0/3 | 0/3 |
| 34 | 2/3 | 0/3 | 3/3 | 0/3 | 0/3 | 0/3 | 0/3 | 0/3 |
| 35 | 2/3 | 0/3 | 3/3 | 0/3 | 0/3 | 0/3 | 0/3 | 0/3 |
| 36 | 3/3 | 2/3 | 3/3 | 2/3 | 0/3 | 0/3 | 0/3 | 3/3 |
| 37 | 0/3 | 1/3 | 0/3 | 1/3 | 0/3 | 0/3 | 0/3 | 0/3 |
| 38 | 3/3 | 0/3 | 1/3 | 0/3 | 0/3 | 0/3 | 0/3 | 0/3 |
| 39 | 1/3 | 2/3 | 3/3 | 2/3 | 3/3 | 0/3 | 0/3 | 0/3 |
| 40 | 1/3 | 0/3 | 3/3 | 1/3 | 0/3 | 0/3 | 0/3 | 1/3 |
| 41 | 0/3 | 0/3 | 0/3 | 0/3 | 0/3 | 0/3 | 0/3 | 0/3 |
| 42 | 1/3 | 0/3 | 0/3 | 1/3 | 0/3 | 0/3 | 0/3 | 0/3 |
| 43 | 2/3 | 0/3 | 1/3 | 0/3 | 0/3 | 0/3 | 0/3 | 0/3 |
| 44 | 0/3 | 0/3 | 0/3 | 0/3 | 0/3 | 0/3 | 0/3 | 1/3 |
| 45 | 2/3 | 0/3 | 0/3 | 0/3 | 0/3 | 0/3 | 0/3 | 0/3 |
| 46 | 2/3 | 0/3 | 0/3 | 0/3 | 0/3 | 0/3 | 0/3 | 0/3 |
| 47 | 3/3 | 3/3 | 0/3 | 2/3 | 0/3 | 0/3 | 0/3 | 2/3 |
| 48 | 1/3 | 0/3 | 0/3 | 0/3 | 0/3 | 0/3 | 0/3 | 0/3 |
| 49 | 1/3 | 0/3 | 2/3 | 0/3 | 0/3 | 0/3 | 0/3 | 0/3 |
| 50 | 1/3 | 0/3 | 2/3 | 0/3 | 0/3 | 0/3 | 0/3 | 1/3 |
| 51 | 0/3 | 0/3 | 0/3 | 3/3 | 0/3 | 0/3 | 0/3 | 0/3 |
| 52 | 1/3 | 0/3 | 0/3 | 0/3 | 0/3 | 0/3 | 0/3 | 1/3 |
| 53 | 3/3 | 0/3 | 3/3 | 0/3 | 0/3 | 0/3 | 0/3 | 3/3 |
| 54 | 3/3 | 0/3 | 1/3 | 0/3 | 0/3 | 0/3 | 0/3 | 0/3 |
| 55 | 0/3 | 0/3 | 0/3 | 1/3 | 0/3 | 0/3 | 0/3 | 0/3 |
| 56 | 2/3 | 0/3 | 0/3 | 0/3 | 0/3 | 0/3 | 0/3 | 0/3 |
| 57 | 2/3 | 0/3 | 3/3 | 2/3 | 0/3 | 0/3 | 0/3 | 0/3 |
| 58 | 0/3 | 0/3 | 0/3 | 0/3 | 0/3 | 0/3 | 0/3 | 0/3 |
| 59 | 2/3 | 0/3 | 1/3 | 0/3 | 0/3 | 0/3 | 0/3 | 0/3 |
| 60 | 1/3 | 0/3 | 0/3 | 0/3 | 0/3 | 0/3 | 0/3 | 0/3 |
| 61 | 1/3 | 0/3 | 2/3 | 3/3 | 0/3 | 0/3 | 0/3 | 3/3 |
| 62 | 2/3 | 0/3 | 0/3 | 0/3 | 0/3 | 0/3 | 0/3 | 0/3 |
| 63 | 0/3 | 0/3 | 0/3 | 0/3 | 0/3 | 0/3 | 0/3 | 0/3 |
| 64 | 3/3 | 0/3 | 0/3 | 0/3 | 0/3 | 0/3 | 0/3 | 0/3 |
| 65 | 1/3 | 0/3 | 0/3 | 0/3 | 0/3 | 0/3 | 0/3 | 0/3 |
| 66 | 3/3 | 0/3 | 0/3 | 0/3 | 0/3 | 0/3 | 0/3 | 0/3 |
| 67 | 1/3 | 1/3 | 1/3 | 2/3 | 2/3 | 0/3 | 0/3 | 1/3 |
| 68 | 1/3 | 0/3 | 0/3 | 0/3 | 0/3 | 0/3 | 0/3 | 1/3 |
| 69 | 0/3 | 0/3 | 0/3 | 2/3 | 3/3 | 0/3 | 0/3 | 1/3 |
| 70 | 3/3 | 0/3 | 1/3 | 0/3 | 0/3 | 1/3 | 0/3 | 1/3 |
| 71 | 3/3 | 0/3 | 1/3 | 1/3 | 0/3 | 0/3 | 0/3 | 0/3 |
| 72 | 0/3 | 0/3 | 0/3 | 0/3 | 0/3 | 0/3 | 0/3 | 0/3 |
| 73 | 0/3 | 0/3 | 0/3 | 0/3 | 0/3 | 0/3 | 0/3 | 0/3 |
| 74 | 0/3 | 0/3 | 0/3 | 0/3 | 0/3 | 0/3 | 0/3 | 0/3 |
| 75 | 3/3 | 0/3 | 0/3 | 2/3 | 0/3 | 0/3 | 0/3 | 1/3 |
| 76 | 0/3 | 0/3 | 0/3 | 0/3 | 0/3 | 0/3 | 0/3 | 0/3 |
| 77 | 3/3 | 0/3 | 0/3 | 0/3 | 0/3 | 0/3 | 0/3 | 0/3 |
| 78 | 1/3 | 0/3 | 0/3 | 0/3 | 0/3 | 0/3 | 0/3 | 0/3 |
| 79 | 2/3 | 0/3 | 0/3 | 0/3 | 0/3 | 0/3 | 0/3 | 0/3 |
| 80 | 0/3 | 0/3 | 0/3 | 0/3 | 0/3 | 0/3 | 0/3 | 0/3 |
| 81 | 3/3 | 0/3 | 0/3 | 0/3 | 0/3 | 0/3 | 0/3 | 0/3 |
| 82 | 0/3 | 0/3 | 0/3 | 0/3 | 0/3 | 0/3 | 0/3 | 0/3 |
| 83 | 3/3 | 0/3 | 2/3 | 2/3 | 0/3 | 0/3 | 0/3 | 0/3 |
| 84 | 3/3 | 0/3 | 0/3 | 0/3 | 0/3 | 0/3 | 0/3 | 0/3 |
| 85 | 3/3 | 2/3 | 0/3 | 0/3 | 0/3 | 0/3 | 0/3 | 0/3 |
| 86 | 3/3 | 3/3 | 0/3 | 3/3 | 0/3 | 0/3 | 0/3 | 0/3 |
| 87 | 0/3 | 0/3 | 0/3 | 0/3 | 0/3 | 0/3 | 0/3 | 0/3 |
| 88 | 0/3 | 0/3 | 3/3 | 0/3 | 0/3 | 0/3 | 0/3 | 0/3 |
| 89 | 0/3 | 3/3 | 3/3 | 0/3 | 0/3 | 0/3 | 0/3 | 0/3 |
| 90 | 0/3 | 0/3 | 0/3 | 0/3 | 0/3 | 0/3 | 0/3 | 0/3 |
| 91 | 1/3 | 0/3 | 1/3 | 0/3 | 0/3 | 0/3 | 0/3 | 0/3 |
| 92 | 0/3 | 0/3 | 2/3 | 0/3 | 0/3 | 0/3 | 0/3 | 0/3 |
| 93 | 1/3 | 0/3 | 0/3 | 0/3 | 0/3 | 0/3 | 0/3 | 0/3 |
| 94 | 3/3 | 2/3 | 2/3 | 2/3 | 2/3 | 0/3 | 3/3 | 1/3 |
| 95 | 3/3 | 3/3 | 3/3 | 2/3 | 0/3 | 0/3 | 3/3 | 2/3 |
| 96 | 3/3 | 3/3 | 3/3 | 3/3 | 0/3 | 3/3 | 0/3 | 3/3 |
| 97 | 3/3 | 3/3 | 3/3 | 2/3 | 0/3 | 0/3 | 0/3 | 1/3 |
| 98 | 3/3 | 3/3 | 3/3 | 3/3 | 3/3 | 3/3 | 3/3 | 3/3 |
| 99 | 0/3 | 0/3 | 0/3 | 0/3 | 0/3 | 0/3 | 0/3 | 0/3 |
| 100 | 3/3 | 3/3 | 3/3 | 2/3 | 0/3 | 3/3 | 3/3 | 3/3 |

outside the bigger shape (*cf.* Figure 9 bottom left). Even though the final decisions of the models were primarily correct, we saw in the answers that except for Claude, none of the models was able to count the shapes correctly.

For the last BP, BP#37 (*cf.* Figure 9 bottom right), the models can identify the concept better, with some even classifying all 12 images correctly (cf. Table 12). The ground truth concept is *triangle vs. circle on top* and for this the perception seems to work more reliably. This could be because the concept of typical shapes positioned above or below one another is generally more familiar compared to other BPs. In this case, the challenge of solving the BP from scratch likely lies more in pattern discovery or reasoning rather than perception.

For completeness, we report image-wise classification results of the qualitative examples of BP#16 (Table 10), BP#29 (Table 11, BP#37 (Table 12) and BP#55 (Table 13).

## B.4. Full vs. Single Images

To examine whether the gap between solving the BP in Task 1 and correctly classifying the underlying concepts in Task 2 arises from how the problem is presented (i.e., as a full image), we conduct an additional analysis using two of the top-performing models from Task 1. This time, we provide the models with the twelve individual images from the BP instead of a single composite image. The prompt remains largely unchanged, with only minor adjustments to the image introduction. We report the results in Table 14. The findings show that this change in setup does not significantly affect the number of correctly solved problems per task, nor does it reduce the observed gap between Task 1 and Task 2.

Table 6: Results of each VLM on the individual Bongard Problems when provided with a selection of 10 possible solutions *(Multiple Choice (10))*. Each model was prompted three times and the number of correct responses is reported (of 3). The reported ratios are marked based on four intervals: less than 1/3 (white), [1/3-2/3) (light green), [2/3-3/3) (green), 3/3 (dark green).

| BP# | o1 | GPT-4o | Claude 3.5 | Gemini 2.0 | Gemini 1.5 | LLaVA-OV | Qwen2VL | InternVL 2.5 |
|---|---|---|---|---|---|---|---|---|
| 1 | 3/3 | 3/3 | 3/3 | 3/3 | 3/3 | 0/3 | 2/3 | 3/3 |
| 2 | 3/3 | 2/3 | 3/3 | 0/3 | 0/3 | 1/3 | 3/3 | 2/3 |
| 3 | 3/3 | 3/3 | 3/3 | 3/3 | 2/3 | 3/3 | 3/3 | 3/3 |
| 4 | 3/3 | 3/3 | 3/3 | 2/3 | 1/3 | 0/3 | 3/3 | 1/3 |
| 5 | 3/3 | 3/3 | 3/3 | 3/3 | 3/3 | 2/3 | 3/3 | 3/3 |
| 6 | 3/3 | 3/3 | 3/3 | 3/3 | 3/3 | 3/3 | 3/3 | 3/3 |
| 7 | 3/3 | 3/3 | 3/3 | 3/3 | 3/3 | 0/3 | 2/3 | 2/3 |
| 8 | 1/3 | 0/3 | 0/3 | 1/3 | 0/3 | 0/3 | 0/3 | 0/3 |
| 9 | 3/3 | 3/3 | 3/3 | 2/3 | 3/3 | 3/3 | 3/3 | 2/3 |
| 10 | 3/3 | 3/3 | 3/3 | 3/3 | 3/3 | 3/3 | 3/3 | 3/3 |
| 11 | 3/3 | 3/3 | 3/3 | 2/3 | 1/3 | 1/3 | 2/3 | 3/3 |
| 12 | 2/3 | 3/3 | 1/3 | 0/3 | 1/3 | 1/3 | 1/3 | 2/3 |
| 13 | 3/3 | 3/3 | 3/3 | 3/3 | 3/3 | 2/3 | 2/3 | 3/3 |
| 14 | 3/3 | 3/3 | 2/3 | 2/3 | 0/3 | 0/3 | 3/3 | 1/3 |
| 15 | 3/3 | 2/3 | 1/3 | 3/3 | 0/3 | 1/3 | 1/3 | 3/3 |
| 16 | 3/3 | 3/3 | 3/3 | 3/3 | 1/3 | 2/3 | 3/3 | 3/3 |
| 17 | 3/3 | 2/3 | 1/3 | 2/3 | 2/3 | 0/3 | 2/3 | 3/3 |
| 18 | 3/3 | 3/3 | 3/3 | 0/3 | 0/3 | 0/3 | 3/3 | 0/3 |
| 19 | 3/3 | 3/3 | 3/3 | 2/3 | 0/3 | 1/3 | 1/3 | 1/3 |
| 20 | 1/3 | 3/3 | 3/3 | 1/3 | 2/3 | 1/3 | 3/3 | 1/3 |
| 21 | 2/3 | 3/3 | 2/3 | 2/3 | 3/3 | 1/3 | 3/3 | 2/3 |
| 22 | 3/3 | 0/3 | 2/3 | 0/3 | 0/3 | 0/3 | 1/3 | 0/3 |
| 23 | 3/3 | 3/3 | 3/3 | 3/3 | 3/3 | 1/3 | 3/3 | 3/3 |
| 24 | 3/3 | 3/3 | 3/3 | 3/3 | 2/3 | 0/3 | 2/3 | 3/3 |
| 25 | 3/3 | 3/3 | 1/3 | 1/3 | 3/3 | 0/3 | 2/3 | 2/3 |
| 26 | 3/3 | 2/3 | 1/3 | 1/3 | 3/3 | 1/3 | 3/3 | 2/3 |
| 27 | 3/3 | 2/3 | 2/3 | 3/3 | 1/3 | 3/3 | 3/3 | 3/3 |
| 28 | 3/3 | 2/3 | 1/3 | 1/3 | 1/3 | 2/3 | 1/3 | 3/3 |
| 29 | 3/3 | 3/3 | 3/3 | 3/3 | 2/3 | 2/3 | 3/3 | 3/3 |
| 30 | 3/3 | 3/3 | 3/3 | 2/3 | 3/3 | 0/3 | 2/3 | 2/3 |
| 31 | 3/3 | 1/3 | 2/3 | 1/3 | 1/3 | 1/3 | 2/3 | 0/3 |
| 32 | 3/3 | 3/3 | 3/3 | 0/3 | 3/3 | 2/3 | 2/3 | 2/3 |
| 33 | 3/3 | 2/3 | 2/3 | 0/3 | 2/3 | 1/3 | 3/3 | 1/3 |
| 34 | 3/3 | 1/3 | 3/3 | 2/3 | 3/3 | 1/3 | 3/3 | 2/3 |
| 35 | 3/3 | 2/3 | 3/3 | 3/3 | 3/3 | 2/3 | 3/3 | 1/3 |
| 36 | 3/3 | 2/3 | 3/3 | 3/3 | 2/3 | 2/3 | 3/3 | 3/3 |
| 37 | 3/3 | 2/3 | 2/3 | 2/3 | 2/3 | 1/3 | 3/3 | 3/3 |
| 38 | 3/3 | 2/3 | 3/3 | 3/3 | 0/3 | 1/3 | 3/3 | 1/3 |
| 39 | 3/3 | 3/3 | 3/3 | 3/3 | 3/3 | 3/3 | 3/3 | 3/3 |
| 40 | 3/3 | 3/3 | 3/3 | 3/3 | 3/3 | 1/3 | 3/3 | 1/3 |
| 41 | 1/3 | 0/3 | 1/3 | 2/3 | 0/3 | 1/3 | 0/3 | 1/3 |
| 42 | 3/3 | 1/3 | 2/3 | 2/3 | 0/3 | 0/3 | 1/3 | 2/3 |
| 43 | 3/3 | 3/3 | 2/3 | 2/3 | 0/3 | 3/3 | 3/3 | 2/3 |
| 44 | 3/3 | 3/3 | 2/3 | 2/3 | 3/3 | 2/3 | 3/3 | 1/3 |
| 45 | 3/3 | 2/3 | 2/3 | 1/3 | 2/3 | 0/3 | 0/3 | 2/3 |
| 46 | 3/3 | 1/3 | 3/3 | 1/3 | 2/3 | 2/3 | 1/3 | 0/3 |
| 47 | 3/3 | 3/3 | 3/3 | 2/3 | 3/3 | 2/3 | 2/3 | 3/3 |
| 48 | 2/3 | 2/3 | 3/3 | 0/3 | 0/3 | 0/3 | 1/3 | 2/3 |
| 49 | 2/3 | 1/3 | 2/3 | 3/3 | 1/3 | 1/3 | 2/3 | 2/3 |
| 50 | 3/3 | 0/3 | 1/3 | 0/3 | 3/3 | 0/3 | 1/3 | 2/3 |

| BP# | o1 | GPT-4o | Claude 3.5 | Gemini 2.0 | Gemini 1.5 | LLaVA-OV | Qwen2VL | InternVL 2.5 |
|---|---|---|---|---|---|---|---|---|
| 51 | 2/3 | 2/3 | 3/3 | 2/3 | 2/3 | 0/3 | 3/3 | 2/3 |
| 52 | 3/3 | 3/3 | 3/3 | 2/3 | 2/3 | 1/3 | 2/3 | 1/3 |
| 53 | 3/3 | 2/3 | 3/3 | 3/3 | 3/3 | 0/3 | 3/3 | 3/3 |
| 54 | 3/3 | 2/3 | 3/3 | 3/3 | 0/3 | 0/3 | 1/3 | 3/3 |
| 55 | 3/3 | 0/3 | 3/3 | 3/3 | 1/3 | 0/3 | 1/3 | 1/3 |
| 56 | 3/3 | 3/3 | 0/3 | 0/3 | 0/3 | 0/3 | 1/3 | 0/3 |
| 57 | 3/3 | 2/3 | 3/3 | 3/3 | 0/3 | 0/3 | 2/3 | 2/3 |
| 58 | 0/3 | 0/3 | 3/3 | 1/3 | 0/3 | 1/3 | 1/3 | 1/3 |
| 59 | 3/3 | 2/3 | 2/3 | 0/3 | 0/3 | 1/3 | 1/3 | 1/3 |
| 60 | 3/3 | 1/3 | 2/3 | 0/3 | 1/3 | 0/3 | 1/3 | 2/3 |
| 61 | 3/3 | 3/3 | 3/3 | 3/3 | 3/3 | 2/3 | 3/3 | 3/3 |
| 62 | 3/3 | 2/3 | 3/3 | 3/3 | 3/3 | 1/3 | 0/3 | 2/3 |
| 63 | 3/3 | 1/3 | 0/3 | 3/3 | 0/3 | 1/3 | 1/3 | 0/3 |
| 64 | 3/3 | 2/3 | 3/3 | 2/3 | 2/3 | 2/3 | 3/3 | 3/3 |
| 65 | 3/3 | 1/3 | 2/3 | 3/3 | 0/3 | 0/3 | 2/3 | 0/3 |
| 66 | 3/3 | 3/3 | 1/3 | 2/3 | 0/3 | 0/3 | 0/3 | 1/3 |
| 67 | 2/3 | 3/3 | 2/3 | 3/3 | 3/3 | 1/3 | 3/3 | 2/3 |
| 68 | 3/3 | 3/3 | 3/3 | 2/3 | 3/3 | 3/3 | 3/3 | 2/3 |
| 69 | 3/3 | 2/3 | 3/3 | 3/3 | 2/3 | 2/3 | 3/3 | 1/3 |
| 70 | 3/3 | 3/3 | 3/3 | 3/3 | 3/3 | 3/3 | 2/3 | 3/3 |
| 71 | 3/3 | 1/3 | 3/3 | 2/3 | 3/3 | 0/3 | 0/3 | 1/3 |
| 72 | 3/3 | 3/3 | 1/3 | 2/3 | 2/3 | 0/3 | 0/3 | 0/3 |
| 73 | 3/3 | 1/3 | 2/3 | 2/3 | 0/3 | 0/3 | 1/3 | 0/3 |
| 74 | 3/3 | 1/3 | 1/3 | 1/3 | 0/3 | 1/3 | 3/3 | 2/3 |
| 75 | 3/3 | 1/3 | 3/3 | 3/3 | 3/3 | 1/3 | 2/3 | 3/3 |
| 76 | 1/3 | 2/3 | 3/3 | 3/3 | 1/3 | 1/3 | 2/3 | 0/3 |
| 77 | 2/3 | 1/3 | 2/3 | 1/3 | 1/3 | 0/3 | 2/3 | 1/3 |
| 78 | 3/3 | 1/3 | 3/3 | 3/3 | 3/3 | 1/3 | 1/3 | 1/3 |
| 79 | 3/3 | 1/3 | 3/3 | 3/3 | 2/3 | 1/3 | 0/3 | 2/3 |
| 80 | 3/3 | 1/3 | 3/3 | 2/3 | 2/3 | 2/3 | 1/3 | 3/3 |
| 81 | 3/3 | 2/3 | 3/3 | 3/3 | 3/3 | 1/3 | 2/3 | 1/3 |
| 82 | 2/3 | 0/3 | 1/3 | 0/3 | 2/3 | 0/3 | 1/3 | 1/3 |
| 83 | 3/3 | 3/3 | 3/3 | 3/3 | 3/3 | 2/3 | 3/3 | 2/3 |
| 84 | 3/3 | 3/3 | 2/3 | 2/3 | 3/3 | 1/3 | 3/3 | 3/3 |
| 85 | 3/3 | 3/3 | 3/3 | 3/3 | 3/3 | 1/3 | 3/3 | 3/3 |
| 86 | 3/3 | 3/3 | 3/3 | 3/3 | 3/3 | 3/3 | 3/3 | 3/3 |
| 87 | 1/3 | 0/3 | 1/3 | 0/3 | 0/3 | 0/3 | 1/3 | 0/3 |
| 88 | 3/3 | 0/3 | 3/3 | 1/3 | 3/3 | 3/3 | 1/3 | 0/3 |
| 89 | 3/3 | 3/3 | 3/3 | 3/3 | 3/3 | 3/3 | 3/3 | 0/3 |
| 90 | 1/3 | 0/3 | 0/3 | 1/3 | 0/3 | 0/3 | 1/3 | 0/3 |
| 91 | 3/3 | 0/3 | 2/3 | 1/3 | 0/3 | 0/3 | 0/3 | 0/3 |
| 92 | 1/3 | 0/3 | 3/3 | 0/3 | 3/3 | 2/3 | 1/3 | 1/3 |
| 93 | 1/3 | 1/3 | 3/3 | 1/3 | 1/3 | 1/3 | 0/3 | 2/3 |
| 94 | 3/3 | 3/3 | 3/3 | 3/3 | 1/3 | 1/3 | 3/3 | 2/3 |
| 95 | 3/3 | 3/3 | 3/3 | 3/3 | 2/3 | 1/3 | 3/3 | 3/3 |
| 96 | 3/3 | 3/3 | 3/3 | 3/3 | 3/3 | 3/3 | 3/3 | 3/3 |
| 97 | 3/3 | 3/3 | 3/3 | 3/3 | 3/3 | 0/3 | 2/3 | 3/3 |
| 98 | 3/3 | 3/3 | 3/3 | 3/3 | 3/3 | 2/3 | 3/3 | 3/3 |
| 99 | 2/3 | 2/3 | 3/3 | 2/3 | 2/3 | 0/3 | 0/3 | 1/3 |
| 100 | 3/3 | 3/3 | 3/3 | 3/3 | 2/3 | 3/3 | 3/3 | 3/3 |

## B.5. Error Types

We provide a full visualization of all results over all tasks and including the different behavior types in Figure 10.

## B.6. Task 3

The detailed results for Task 3 are reported in Table 15 and Table 16.

## B.7. Especially Hard Bongard Problems

Throughout the experiments, we saw that the investigated VLMs can solve some BPs better than others, where the set of solved puzzles appears to differ across task setting. Interestingly, we find that there are 10 BPs that are especially hard, *i.e.*, they were not solved in any of the tasks. These are mainly BPs categorized as targeting *spatial* relations, such as points inside figure are on a line or not (BP#42). But also BPs from other categories are part of it, such as BP#82 that asks for the convex hull of crosses forming an equilateral triangle (concept) and BP#58 where the solid squares are same size/not same size (same). In Figure 11 each of the 10 BPs is depicted.

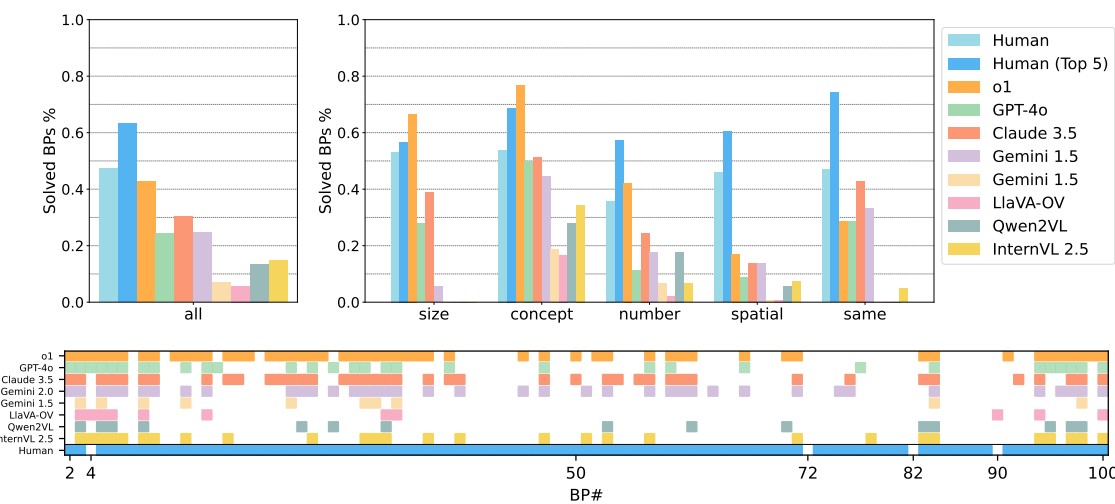

Figure 8: **Human results compared to results of VLMs.** Mean results across all BPs (top left) together with mean results in different categories (top right). Number of BPs solved at least once for different models (bottom).

Table 7: Mean percentages of solved BPs for humans and VLMs across categories.

|         | Human | Human (Top 5) | o1    | GPT-4o | Claude 3.5 | Gemini 2.0 | Gemini 1.5 | LlaVA-OV | Qwen2VL | InternVL 2.5 |
|---------|-------|---------------|-------|--------|------------|------------|------------|----------|---------|--------------|
| size    | 53.17 | 56.67         | 66.67 | 27.78  | 38.89      | 5.56       | 0.00       | 0.00     | 0.00    | 0.00         |
| concept | 53.81 | 68.67         | 76.67 | 50.00  | 51.11      | 44.44      | 18.89      | 16.67    | 27.78   | 34.44        |
| number  | 35.87 | 57.33         | 42.22 | 11.11  | 24.44      | 17.78      | 6.67       | 2.22     | 17.78   | 6.67         |
| spatial | 45.88 | 60.49         | 17.07 | 8.94   | 13.82      | 13.82      | 0.81       | 0.81     | 5.69    | 7.32         |
| same    | 46.94 | 74.29         | 28.57 | 28.57  | 42.86      | 33.33      | 0.00       | 0.00     | 0.00    | 4.76         |
| all     | 47.28 | 63.23         | 42.76 | 24.24  | 30.30      | 24.58      | 7.07       | 5.72     | 13.47   | 14.81        |

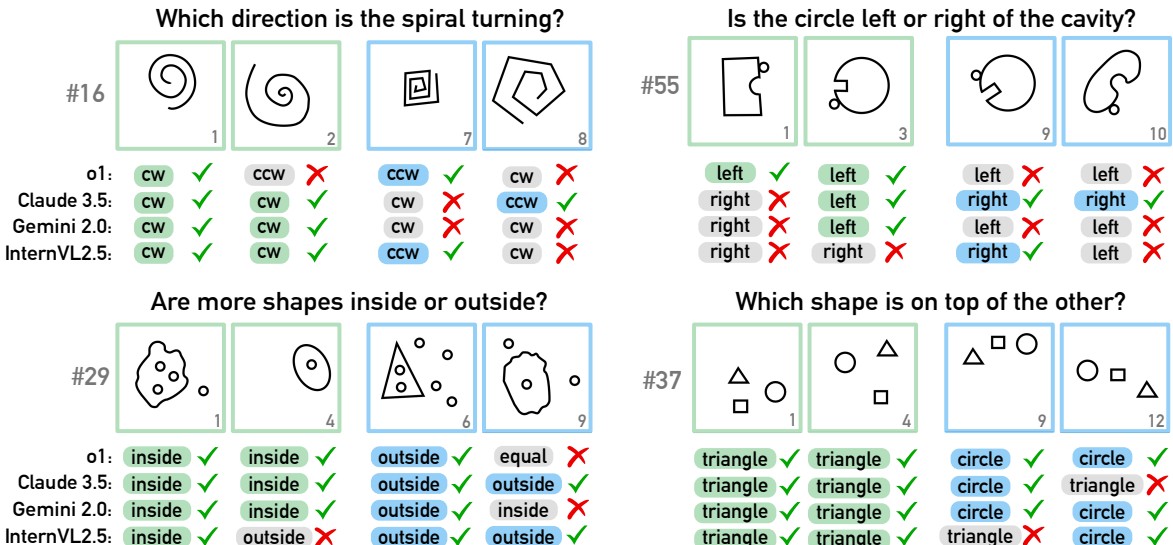

Figure 9: **Task 2: VLMs fail to identify simple visual concepts.** VLMs are challenged with identifying visual concepts in BPs. Although the VLM is able to recognize some of the concepts when specifically asked for (bottom), on the others, it continues to falter (top). Abbreviations used for *clockwise (cw)* and *counter-clockwise (ccw)*.

Table 8: Accuracy of each VLM when classifying the BP images in Task 2.

| BP# | o1 | GPT-4o | Claude 3.5 | Gemini 2.0 | Gemini 1.5 | LlaVA-OV | Qwen2VL | InternVL 2.5 | BP# | o1 | GPT-4o | Claude 3.5 | Gemini 2.0 | Gemini 1.5 | LlaVA-OV | Qwen2VL | InternVL 2.5 |
|---|---|---|---|---|---|---|---|---|---|---|---|---|---|---|---|---|---|
| 1 | 66.67 | 100.00 | 100.00 | 97.22 | 100.00 | 100.00 | 100.00 | 97.22 | 51 | 83.33 | 83.33 | 72.22 | 72.22 | 86.11 | 83.33 | 58.33 | 80.56 |
| 2 | 75.00 | 100.00 | 80.56 | 100.00 | 100.00 | 86.11 | 75.00 | 61.11 | 52 | 58.33 | 47.22 | 55.56 | 38.89 | 61.11 | 52.78 | 66.67 | 52.78 |
| 3 | 100.00 | 100.00 | 83.33 | 100.00 | 100.00 | 100.00 | 100.00 | 100.00 | 53 | 83.33 | 77.78 | 91.67 | 100.00 | 88.89 | 72.22 | 66.67 | 83.33 |
| 4 | 100.00 | 100.00 | 100.00 | 94.44 | 91.67 | 69.44 | 100.00 | 88.89 | 54 | 33.33 | 50.00 | 50.00 | 52.78 | 41.67 | 47.22 | 41.67 | 38.89 |
| 5 | 100.00 | 100.00 | 100.00 | 100.00 | 100.00 | 100.00 | 100.00 | 100.00 | 55 | 50.00 | 50.00 | 50.00 | 41.67 | 41.67 | 50.00 | 50.00 | 41.67 |
| 6 | 100.00 | 100.00 | 86.11 | 83.33 | 83.33 | 83.33 | 91.67 | 91.67 | 56 | 91.67 | 91.67 | 66.67 | 75.00 | 91.67 | 50.00 | 58.33 | 75.00 |
| 7 | 100.00 | 100.00 | 94.44 | 100.00 | 100.00 | 100.00 | 100.00 | 100.00 | 57 | 66.67 | 100.00 | 97.22 | 69.44 | 58.33 | 91.67 | 100.00 | 83.33 |
| 8 | 100.00 | 100.00 | 94.44 | 100.00 | 100.00 | 100.00 | 100.00 | 100.00 | 58 | 75.00 | 66.67 | 88.89 | 94.44 | 61.11 | 83.33 | 75.00 | 72.22 |
| 9 | 91.67 | 86.11 | 77.78 | 86.11 | 91.67 | 88.89 | 91.67 | 88.89 | 59 | 83.33 | 86.11 | 88.89 | 69.44 | 88.89 | 91.67 | 75.00 | 80.56 |
| 10 | 83.33 | 83.33 | 72.22 | 72.22 | 83.33 | 94.44 | 91.67 | 83.33 | 60 | 91.67 | 75.00 | 97.22 | 91.67 | 91.67 | 88.89 | 75.00 | 80.56 |
| 11 | 91.67 | 91.67 | 100.00 | 88.89 | 91.67 | 75.00 | 91.67 | 83.33 | 61 | 50.00 | 41.67 | 44.44 | 41.67 | 44.44 | 50.00 | 41.67 | 38.89 |
| 12 | 100.00 | 100.00 | 91.67 | 97.22 | 97.22 | 80.56 | 66.67 | 75.00 | 62 | 100.00 | 75.00 | 69.44 | 72.22 | 58.33 | 80.56 | 75.00 | 69.44 |
| 13 | 100.00 | 83.33 | 94.44 | 61.11 | 66.67 | 50.00 | 50.00 | 44.44 | 63 | 83.33 | 75.00 | 97.22 | 94.44 | 100.00 | 88.89 | 100.00 | 61.11 |
| 14 | 50.00 | 77.78 | 58.33 | 55.56 | 61.11 | 83.33 | 50.00 | 47.22 | 64 | 58.33 | 38.89 | 38.89 | 52.78 | 52.78 | 66.67 | 50.00 | 41.67 |
| 15 | 91.67 | 88.89 | 83.33 | 100.00 | 72.22 | 50.00 | 75.00 | 75.00 | 65 | 58.33 | 50.00 | 77.78 | 77.78 | 72.22 | 63.89 | 75.00 | 58.33 |
| 16 | 50.00 | 50.00 | 44.44 | 52.78 | 50.00 | 63.89 | 50.00 | 50.00 | 66 | 83.33 | 83.33 | 77.78 | 52.78 | 25.00 | 63.89 | 58.33 | 55.56 |
| 17 | 66.67 | 63.89 | 80.56 | 80.56 | 72.22 | 72.22 | 75.00 | 63.89 | 67 | 58.33 | 58.33 | 41.67 | 69.44 | 69.44 | 52.78 | 41.67 | 52.78 |
| 18 | 50.00 | 50.00 | 38.89 | 61.11 | 50.00 | 52.78 | 50.00 | 55.56 | 68 | 66.67 | 58.33 | 91.67 | 63.89 | 86.11 | 72.22 | 75.00 | 75.00 |
| 19 | 83.33 | 80.56 | 72.22 | 97.22 | 91.67 | 75.00 | 66.67 | 63.89 | 69 | 58.33 | 50.00 | 58.33 | 75.00 | 72.22 | 66.67 | 58.33 | 75.00 |
| 20 | 58.33 | 52.78 | 58.33 | 55.56 | 55.56 | 50.00 | 50.00 | 61.11 | 70 | 50.00 | 66.67 | 44.44 | 41.67 | 44.44 | 50.00 | 58.33 | 52.78 |
| 21 | 66.67 | 58.33 | 66.67 | 44.44 | 100.00 | 83.33 | 100.00 | 97.22 | 71 | 100.00 | 88.89 | 75.00 | 94.44 | 88.89 | 77.78 | 66.67 | 66.67 |
| 22 | 75.00 | 77.78 | 77.78 | 66.67 | 58.33 | 97.22 | 91.67 | 52.78 | 72 | 100.00 | 83.33 | 77.78 | 83.33 | 55.56 | 75.00 | 75.00 | 47.22 |
| 23 | 91.67 | 97.22 | 100.00 | 100.00 | 100.00 | 100.00 | 100.00 | 100.00 | 73 | 83.33 | 66.67 | 72.22 | 58.33 | 77.78 | 61.11 | 66.67 | 41.67 |
| 24 | 100.00 | 100.00 | 100.00 | 100.00 | 100.00 | 100.00 | 100.00 | 100.00 | 74 | 66.67 | 47.22 | 47.22 | 58.33 | 61.11 | 44.44 | 41.67 | 47.22 |
| 25 | 100.00 | 100.00 | 100.00 | 100.00 | 100.00 | 91.67 | 83.33 | 100.00 | 75 | 50.00 | 47.22 | 50.00 | 38.89 | 50.00 | 41.67 | 50.00 | 55.56 |
| 26 | 100.00 | 100.00 | 100.00 | 100.00 | 75.00 | 77.78 | 91.67 | 77.78 | 76 | 75.00 | 50.00 | 50.00 | 72.22 | 75.00 | 38.89 | 50.00 | 47.22 |
| 27 | 100.00 | 100.00 | 91.67 | 94.44 | 88.89 | 66.67 | 91.67 | 80.56 | 77 | 50.00 | 47.22 | 66.67 | 58.33 | 50.00 | 77.78 | 41.67 | 63.89 |
| 28 | 91.67 | 91.67 | 100.00 | 97.22 | 100.00 | 91.67 | 91.67 | 94.44 | 78 | 50.00 | 50.00 | 44.44 | 52.78 | 50.00 | 47.22 | 58.33 | 47.22 |
| 29 | 91.67 | 75.00 | 100.00 | 94.44 | 91.67 | 88.89 | 91.67 | 75.00 | 79 | 100.00 | 50.00 | 91.67 | 94.44 | 100.00 | 77.78 | 100.00 | 86.11 |
| 30 | 91.67 | 66.67 | 61.11 | 77.78 | 83.33 | 91.67 | 83.33 | 58.33 | 80 | 50.00 | 50.00 | 52.78 | 50.00 | 50.00 | 52.78 | 50.00 | 66.67 |
| 31 | 75.00 | 75.00 | 75.00 | 77.78 | 66.67 | 80.56 | 75.00 | 72.22 | 81 | 91.67 | 69.44 | 72.22 | 69.44 | 58.33 | 50.00 | 50.00 | 77.78 |
| 32 | 66.67 | 83.33 | 69.44 | 75.00 | 83.33 | 75.00 | 83.33 | 63.89 | 82 | 50.00 | 58.33 | 55.56 | 58.33 | 58.33 | 61.11 | 75.00 | 50.00 |
| 33 | 83.33 | 91.67 | 91.67 | 86.11 | 77.78 | 69.44 | 83.33 | 86.11 | 83 | 75.00 | 75.00 | 80.56 | 83.33 | 83.33 | 72.22 | 75.00 | 69.44 |
| 34 | 66.67 | 91.67 | 80.56 | 91.67 | 91.67 | 75.00 | 66.67 | 72.22 | 84 | 91.67 | 83.33 | 80.56 | 83.33 | 83.33 | 80.56 | 75.00 | 77.78 |
| 35 | 83.33 | 75.00 | 47.22 | 61.11 | 63.89 | 41.67 | 58.33 | 52.78 | 85 | 66.67 | 61.11 | 77.78 | 55.56 | 83.33 | 58.33 | 75.00 | 47.22 |
| 36 | 100.00 | 100.00 | 100.00 | 100.00 | 100.00 | 100.00 | 100.00 | 100.00 | 86 | 75.00 | 97.22 | 77.78 | 27.78 | 83.33 | 83.33 | 75.00 | 44.44 |
| 37 | 100.00 | 100.00 | 91.67 | 97.22 | 100.00 | 91.67 | 91.67 | 88.89 | 87 | 75.00 | 83.33 | 61.11 | 52.78 | 75.00 | 30.56 | 50.00 | 50.00 |
| 38 | 100.00 | 97.22 | 94.44 | 100.00 | 100.00 | 97.22 | 100.00 | 36.11 | 88 | 91.67 | 80.56 | 91.67 | 86.11 | 91.67 | 86.11 | 83.33 | 61.11 |
| 39 | 100.00 | 91.67 | 86.11 | 66.67 | 83.33 | 69.44 | 100.00 | 72.22 | 89 | 75.00 | 75.00 | 97.22 | 100.00 | 88.89 | 88.89 | 58.33 | 83.33 |
| 40 | 58.33 | 55.56 | 58.33 | 66.67 | 55.56 | 50.00 | 50.00 | 66.67 | 90 | 75.00 | 52.78 | 66.67 | 55.56 | 55.56 | 58.33 | 58.33 | 50.00 |
| 41 | 50.00 | 50.00 | 66.67 | 58.33 | 58.33 | 50.00 | 50.00 | 50.00 | 91 | 75.00 | 66.67 | 83.33 | 83.33 | 86.11 | 77.78 | 66.67 | 69.44 |
| 42 | 50.00 | 50.00 | 52.78 | 61.11 | 86.11 | 50.00 | 50.00 | 63.89 | 92 | 66.67 | 66.67 | 41.67 | 61.11 | 91.67 | 58.33 | 75.00 | 52.78 |
| 43 | 83.33 | 83.33 | 94.44 | 100.00 | 83.33 | 75.00 | 83.33 | 83.33 | 93 | 41.67 | 61.11 | 61.11 | 63.89 | 75.00 | 52.78 | 50.00 | 63.89 |
| 44 | 58.33 | 50.89 | 91.67 | 75.00 | 58.33 | 50.00 | 66.67 | 63.89 | 94 | 100.00 | 100.00 | 100.00 | 100.00 | 97.22 | 100.00 | 100.00 | 97.22 |
| 45 | 66.67 | 58.33 | 63.89 | 69.44 | 33.33 | 50.00 | 58.33 | 52.78 | 95 | 100.00 | 100.00 | 100.00 | 100.00 | 100.00 | 100.00 | 100.00 | 100.00 |
| 46 | 75.00 | 66.67 | 83.33 | 80.56 | 55.56 | 63.89 | 75.00 | 72.22 | 96 | 100.00 | 100.00 | 44.44 | 77.78 | 52.78 | 63.89 | 75.00 | 63.89 |
| 47 | 91.67 | 83.33 | 83.33 | 91.67 | 77.78 | 58.33 | 75.00 | 77.78 | 97 | 100.00 | 97.22 | 97.22 | 97.22 | 100.00 | 100.00 | 100.00 | 100.00 |
| 48 | 91.67 | 94.44 | 100.00 | 100.00 | 100.00 | 61.11 | 100.00 | 94.44 | 98 | 66.67 | 91.67 | 100.00 | 91.67 | 91.67 | 94.44 | 91.67 | 94.44 |
| 49 | 100.00 | 63.89 | 80.56 | 55.56 | 58.33 | 55.56 | 41.67 | 55.56 | 99 | 50.00 | 50.00 | 55.56 | 52.78 | 63.89 | 69.44 | 58.33 | 55.56 |
| 50 | 100.00 | 77.78 | 94.44 | 88.89 | 91.67 | 83.33 | 91.67 | 80.56 | 100 | 91.67 | 83.33 | 100.00 | 100.00 | 100.00 | 94.44 | 100.00 | 94.44 |

Table 9: **The number of visual patterns that the VLMs can correctly identify is low (Task 2).** Number of concepts of the single BPs that the VLMs were able to classify correctly for the 12 images of the BP. BPs either counted when for all 12 images 3/3 times the model was correct or when it was at least 2/3 times correct for each image.

| Model | Task 2 All 3/3 | Task 2 All 2/3 | \| T1 ∩ T2 \| (Both 2/3) | \| T1 ∩ T2 \| / T1 (Both 2/3) |
|---|---|---|---|---|
| o1 | **24** | 24 | 17 | 31.48% |
| GPT-4o | 19 | 24 | 15 | **46.88%** |
| Claude | 16 | 22 | 15 | 34.88% |
| Gemini 2.0 Flash | 19 | **30** | **18** | 45.00% |
| Gemini 1.5 Pro | 20 | 22 | 4 | 36.36% |
| LLaVA-OneVision | 11 | 15 | 4 | 33.33% |
| Qwen2VL | 20 | 20 | 6 | 37.50% |
| InternVL2.5 | 10 | 16 | 10 | 34.48% |

Table 10: **BP#16.** Classification results when providing the single images of BP#16 and asking for clockwise or counter-clockwise.

| | Clockwise | | | | | | Counter-clockwise | | | | | |
|---|---|---|---|---|---|---|---|---|---|---|---|---|
| | 1 | 2 | 3 | 4 | 5 | 6 | 7 | 8 | 9 | 10 | 11 | 12 |
| o1 | 1/1 | 0/1 | 0/1 | 0/1 | 1/1 | 1/1 | 1/1 | 0/1 | 1/1 | 0/1 | 1/1 | 0/1 |
| GPT-4o | 3/3 | 3/3 | 3/3 | 3/3 | 3/3 | 3/3 | 0/3 | 0/3 | 0/3 | 0/3 | 0/3 | 0/3 |
| Claude 3.5 | 2/3 | 3/3 | 1/3 | 2/3 | 3/3 | 2/3 | 0/3 | 1/3 | 0/3 | 1/3 | 0/3 | 1/3 |
| Gemini 2.0 | 2/3 | 3/3 | 2/3 | 3/3 | 3/3 | 3/3 | 0/3 | 2/3 | 0/3 | 0/3 | 1/3 | 0/3 |
| Gemini 1.5 | 3/3 | 3/3 | 2/3 | 2/3 | 3/3 | 3/3 | 0/3 | 0/3 | 0/3 | 0/3 | 2/3 | 0/3 |
| Llava-Onevision | 2/3 | 2/3 | 2/3 | 2/3 | 1/3 | 2/3 | 3/3 | 2/3 | 1/3 | 3/3 | 1/3 | 2/3 |
| Qwen2VL | 3/3 | 3/3 | 3/3 | 3/3 | 3/3 | 3/3 | 0/3 | 0/3 | 0/3 | 0/3 | 0/3 | 0/3 |
| InternVL2.5 | 2/3 | 2/3 | 1/3 | 1/3 | 2/3 | 1/3 | 3/3 | 1/3 | 2/3 | 1/3 | 1/3 | 1/3 |

Table 11: **BP#29.** Correctly classified concepts of BP#29. Models were asked wether there are more shapes inside or outside the big figure.

| | Inside | | | | | | Outside | | | | | |
|---|---|---|---|---|---|---|---|---|---|---|---|---|
| | 1 | 2 | 3 | 4 | 5 | 6 | 7 | 8 | 9 | 10 | 11 | 12 |
| o1 | 1/1 | 1/1 | 1/1 | 1/1 | 1/1 | 1/1 | 1/1 | 1/1 | 0/1 | 1/1 | 1/1 | 1/1 |
| GPT-4o | 3/3 | 3/3 | 3/3 | 3/3 | 3/3 | 3/3 | 3/3 | 0/3 | 0/3 | 1/3 | 2/3 | 3/3 |
| Claude 3.5 | 3/3 | 3/3 | 3/3 | 3/3 | 3/3 | 3/3 | 3/3 | 3/3 | 3/3 | 3/3 | 3/3 | 3/3 |
| Gemini 2.0 | 3/3 | 3/3 | 3/3 | 3/3 | 3/3 | 3/3 | 3/3 | 3/3 | 1/3 | 3/3 | 3/3 | 3/3 |
| Gemini 1.5 | 3/3 | 3/3 | 3/3 | 3/3 | 3/3 | 0/3 | 3/3 | 3/3 | 3/3 | 3/3 | 3/3 | 3/3 |
| Llava-Onevision | 3/3 | 3/3 | 2/3 | 3/3 | 3/3 | 3/3 | 3/3 | 2/3 | 1/3 | 3/3 | 3/3 | 3/3 |
| Qwen2VL | 3/3 | 3/3 | 3/3 | 3/3 | 3/3 | 3/3 | 3/3 | 3/3 | 0/3 | 3/3 | 3/3 | 3/3 |
| InternVL2.5 | 3/3 | 3/3 | 1/3 | 0/3 | 2/3 | 2/3 | 2/3 | 2/3 | 3/3 | 3/3 | 3/3 | 3/3 |

Table 12: **BP#37.** Correctly classified concepts for BP#37. Models were asked to output whether triangle or circle is on top.

| | Triangle | | | | | | Circle | | | | | |
|---|---|---|---|---|---|---|---|---|---|---|---|---|
| | 1 | 2 | 3 | 4 | 5 | 6 | 7 | 8 | 9 | 10 | 11 | 12 |
| o1 | 1/1 | 1/1 | 1/1 | 1/1 | 1/1 | 1/1 | 1/1 | 1/1 | 1/1 | 1/1 | 1/1 | 1/1 |
| GPT-4o | 3/3 | 3/3 | 3/3 | 3/3 | 3/3 | 3/3 | 3/3 | 3/3 | 3/3 | 3/3 | 3/3 | 3/3 |
| Claude 3.5 | 3/3 | 3/3 | 3/3 | 3/3 | 3/3 | 3/3 | 3/3 | 3/3 | 3/3 | 3/3 | 3/3 | 0/3 |
| Gemini 2.0 | 3/3 | 3/3 | 3/3 | 3/3 | 3/3 | 3/3 | 3/3 | 3/3 | 3/3 | 3/3 | 3/3 | 2/3 |
| Gemini 1.5 | 3/3 | 3/3 | 3/3 | 3/3 | 3/3 | 3/3 | 3/3 | 3/3 | 3/3 | 3/3 | 3/3 | 3/3 |
| Llava-Onevision | 3/3 | 3/3 | 3/3 | 3/3 | 3/3 | 3/3 | 3/3 | 3/3 | 0/3 | 3/3 | 3/3 | 3/3 |
| Qwen2VL | 3/3 | 3/3 | 3/3 | 3/3 | 3/3 | 3/3 | 3/3 | 3/3 | 0/3 | 3/3 | 3/3 | 3/3 |
| InternVL2.5 | 3/3 | 2/3 | 3/3 | 2/3 | 3/3 | 3/3 | 3/3 | 3/3 | 1/3 | 3/3 | 3/3 | 3/3 |

Table 13: **BP#55.** Correctly classified for concepts of BP#55. Models were asked whether the circle shape is located left or right from the cavity in the big shape (viewed from inside the shape).

| | Left | | | | | | Right | | | | | |
|---|---|---|---|---|---|---|---|---|---|---|---|---|
| | 1 | 2 | 3 | 4 | 5 | 6 | 7 | 8 | 9 | 10 | 11 | 12 |
| o1 | 1/1 | 1/1 | 1/1 | 1/1 | 0/1 | 0/1 | 1/1 | 0/1 | 0/1 | 0/1 | 0/1 | 1/1 |
| GPT-4o | 0/3 | 3/3 | 3/3 | 3/3 | 0/3 | 0/3 | 3/3 | 0/3 | 0/3 | 3/3 | 0/3 | 3/3 |
| Claude 3.5 | 1/3 | 2/3 | 2/3 | 2/3 | 0/3 | 0/3 | 2/3 | 0/3 | 2/3 | 3/3 | 1/3 | 3/3 |
| Gemini 2.0 | 0/3 | 3/3 | 3/3 | 3/3 | 0/3 | 0/3 | 3/3 | 0/3 | 0/3 | 0/3 | 0/3 | 3/3 |
| Gemini 1.5 | 0/3 | 3/3 | 3/3 | 3/3 | 0/3 | 0/3 | 3/3 | 0/3 | 0/3 | 0/3 | 0/3 | 3/3 |
| Llava-Onevision | 0/3 | 3/3 | 3/3 | 3/3 | 0/3 | 0/3 | 3/3 | 0/3 | 0/3 | 3/3 | 0/3 | 3/3 |
| Qwen2VL | 0/3 | 3/3 | 3/3 | 3/3 | 0/3 | 0/3 | 3/3 | 0/3 | 0/3 | 3/3 | 0/3 | 3/3 |
| InternVL2.5 | 0/3 | 3/3 | 1/3 | 3/3 | 0/3 | 0/3 | 3/3 | 0/3 | 2/3 | 1/3 | 0/3 | 2/3 |

Table 14: Comparison of model performance on Task 1 and 2 with original setup (full image) and single images as input.

| | T1 w/o T2 | T2 | T2 w/o T1 |
|---|---|---|---|
| GPT-4o (orig) | 13 | 11 | 13 |
| GPT-4o (single images) | 12 | 12 | 12 |
| Claude (orig) | 20 | 11 | 11 |
| Claude (single images) | 20 | 12 | 10 |

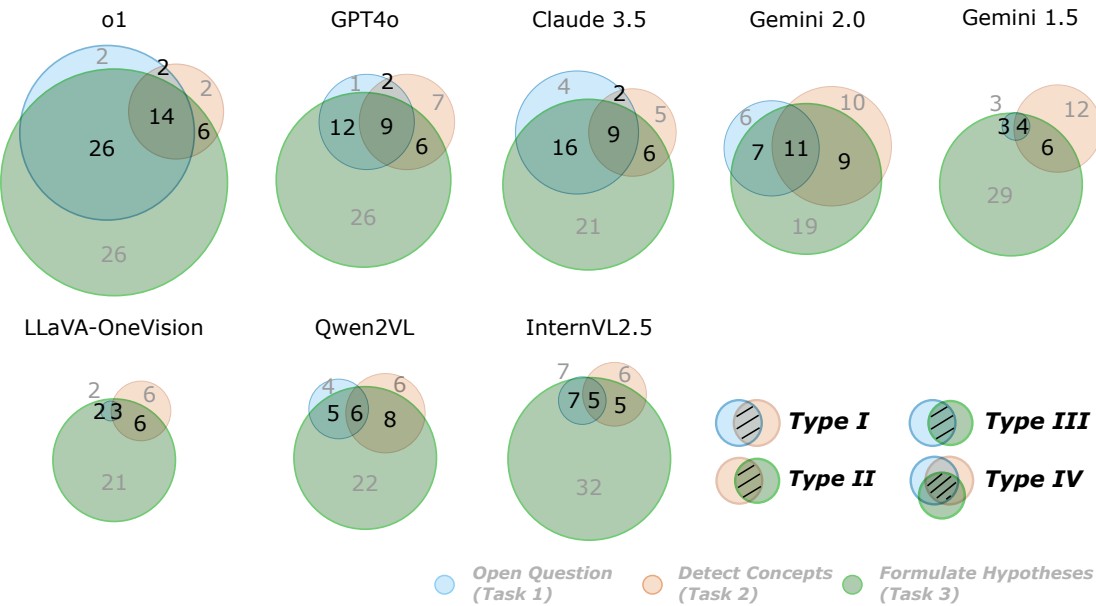

Figure 10: Solved BPs across all tasks, Task 1, Task 2 and Task 3, further highlighting the intersections of these, Type I, Type II, Type III and Type IV.

Table 15: Task 3: Number of correct hypotheses together with number of BPs for specific types.

| Model | Task 3 | $|T1 \cap T3|$ / T1 | Type I | Type II | Type III | Type IV | Type IV/ T1 |
|---|---|---|---|---|---|---|---|
| o1 | **72** | 90.91% | 2 | 6 | 26 | 14 | 31.82% |
| GPT-4o | 53 | 87.50% | 2 | 6 | 12 | 9 | 37.50% |
| Claude | 52 | 80.65% | 2 | 6 | 16 | 9 | 29.03% |
| Gemini 2.0 Flash | 46 | 75.00% | 0 | 9 | 7 | 11 | 45.83% |
| Gemini 1.5 Pro | 42 | **100.00%** | 0 | 6 | 6 | 4 | 57.14% |
| LLaVA Onevision | 32 | **100.00%** | 0 | 6 | 3 | 4 | 60.00% |
| Qwen2VL | 41 | 73.33% | 0 | 8 | 5 | 6 | 40.00% |
| InternVL2.5 | 49 | **100.00%** | 0 | 5 | 7 | 5 | 41.67% |

Table 16: **Results of each VLM on Task 3.** Each model was prompted once to generate 20 hypotheses. The table reports whether a correct hypotheses was included in the response (1) or not (0).

| BP# | o1 | GPT-4o | Claude 3.5 | Gemini 2.0 | Gemini 1.5 | LlaVA-OV | Qwen2VL | InternVL 2.5 |
|---|---|---|---|---|---|---|---|---|
| 1 | 0 | 1 | 1 | 1 | 1 | 1 | 1 | 1 |
| 2 | 1 | 0 | 1 | 0 | 0 | 1 | 0 | 1 |
| 3 | 1 | 1 | 1 | 1 | 1 | 1 | 1 | 1 |
| 4 | 1 | 0 | 1 | 1 | 1 | 0 | 1 | 1 |
| 5 | 1 | 1 | 1 | 1 | 1 | 1 | 1 | 1 |
| 6 | 1 | 1 | 1 | 1 | 1 | 1 | 1 | 1 |
| 7 | 1 | 1 | 1 | 1 | 0 | 1 | 1 | 1 |
| 8 | 0 | 0 | 0 | 0 | 0 | 0 | 0 | 0 |
| 9 | 1 | 1 | 1 | 1 | 1 | 1 | 1 | 1 |
| 10 | 1 | 1 | 0 | 1 | 1 | 0 | 1 | 1 |
| 11 | 1 | 1 | 1 | 1 | 1 | 0 | 0 | 1 |
| 12 | 1 | 1 | 0 | 1 | 0 | 1 | 0 | 1 |
| 13 | 1 | 1 | 0 | 1 | 0 | 1 | 0 | 1 |
| 14 | 1 | 0 | 1 | 1 | 1 | 1 | 0 | 0 |
| 15 | 1 | 1 | 1 | 1 | 1 | 1 | 1 | 1 |
| 16 | 0 | 0 | 0 | 0 | 0 | 0 | 0 | 0 |
| 17 | 1 | 1 | 1 | 1 | 1 | 1 | 1 | 1 |
| 18 | 1 | 1 | 1 | 1 | 0 | 0 | 0 | 1 |
| 19 | 1 | 1 | 0 | 1 | 1 | 1 | 1 | 1 |
| 20 | 1 | 0 | 1 | 0 | 0 | 0 | 0 | 0 |
| 21 | 0 | 1 | 1 | 0 | 1 | 1 | 1 | 0 |
| 22 | 0 | 0 | 1 | 0 | 0 | 1 | 0 | 0 |
| 23 | 1 | 1 | 0 | 1 | 1 | 1 | 1 | 1 |
| 24 | 0 | 1 | 1 | 1 | 0 | 1 | 1 | 1 |
| 25 | 1 | 0 | 1 | 0 | 0 | 1 | 1 | 0 |
| 26 | 0 | 1 | 0 | 0 | 0 | 0 | 0 | 0 |
| 27 | 1 | 0 | 1 | 1 | 0 | 0 | 1 | 1 |
| 28 | 1 | 1 | 1 | 0 | 1 | 0 | 1 | 1 |
| 29 | 1 | 1 | 0 | 1 | 0 | 0 | 0 | 1 |
| 30 | 1 | 1 | 1 | 1 | 1 | 0 | 0 | 1 |
| 31 | 1 | 1 | 1 | 1 | 1 | 1 | 1 | 1 |
| 32 | 1 | 1 | 1 | 1 | 1 | 1 | 1 | 1 |
| 33 | 1 | 1 | 1 | 1 | 1 | 1 | 0 | 1 |
| 34 | 1 | 0 | 1 | 0 | 1 | 0 | 0 | 0 |
| 35 | 1 | 0 | 1 | 0 | 0 | 0 | 0 | 1 |
| 36 | 1 | 1 | 1 | 1 | 0 | 0 | 0 | 1 |
| 37 | 1 | 0 | 1 | 1 | 1 | 0 | 0 | 0 |
| 38 | 1 | 1 | 1 | 1 | 0 | 0 | 1 | 1 |
| 39 | 1 | 0 | 1 | 0 | 1 | 0 | 1 | 1 |
| 40 | 0 | 1 | 0 | 0 | 0 | 0 | 0 | 0 |
| 41 | 1 | 0 | 0 | 0 | 0 | 1 | 0 | 0 |
| 42 | 1 | 0 | 0 | 0 | 0 | 0 | 0 | 0 |
| 43 | 0 | 1 | 0 | 0 | 0 | 0 | 0 | 0 |
| 44 | 1 | 0 | 1 | 0 | 0 | 0 | 0 | 0 |
| 45 | 0 | 0 | 1 | 0 | 0 | 0 | 0 | 0 |
| 46 | 1 | 0 | 0 | 1 | 0 | 0 | 0 | 0 |
| 47 | 1 | 1 | 0 | 0 | 0 | 0 | 0 | 1 |
| 48 | 0 | 0 | 1 | 0 | 0 | 0 | 0 | 0 |
| 49 | 1 | 0 | 0 | 0 | 1 | 0 | 1 | 0 |
| 50 | 1 | 1 | 1 | 1 | 1 | 1 | 1 | 1 |
| 51 | 0 | 1 | 1 | 1 | 0 | 0 | 1 | 1 |
| 52 | 1 | 1 | 1 | 0 | 1 | 0 | 1 | 0 |
| 53 | 1 | 0 | 0 | 1 | 0 | 0 | 0 | 0 |
| 54 | 1 | 0 | 0 | 0 | 0 | 0 | 0 | 0 |
| 55 | 0 | 0 | 0 | 0 | 0 | 0 | 0 | 1 |
| 56 | 0 | 0 | 0 | 0 | 0 | 1 | 0 | 0 |
| 57 | 1 | 1 | 1 | 0 | 1 | 1 | 1 | 0 |
| 58 | 0 | 0 | 0 | 0 | 0 | 0 | 0 | 0 |
| 59 | 1 | 0 | 0 | 0 | 1 | 0 | 0 | 1 |
| 60 | 1 | 0 | 1 | 1 | 0 | 1 | 1 | 0 |
| 61 | 1 | 1 | 1 | 1 | 0 | 1 | 0 | 1 |
| 62 | 1 | 0 | 0 | 0 | 1 | 0 | 0 | 1 |
| 63 | 0 | 0 | 0 | 0 | 0 | 0 | 0 | 0 |
| 64 | 0 | 0 | 0 | 0 | 0 | 0 | 0 | 0 |
| 65 | 0 | 0 | 0 | 0 | 0 | 0 | 0 | 0 |
| 66 | 1 | 1 | 0 | 0 | 0 | 0 | 0 | 0 |
| 67 | 1 | 0 | 0 | 0 | 1 | 0 | 1 | 0 |
| 68 | 0 | 1 | 0 | 1 | 0 | 0 | 1 | 1 |
| 69 | 1 | 0 | 0 | 1 | 0 | 0 | 0 | 0 |
| 70 | 1 | 0 | 1 | 0 | 1 | 0 | 0 | 0 |
| 71 | 1 | 1 | 1 | 1 | 1 | 1 | 1 | 1 |
| 72 | 1 | 0 | 0 | 0 | 0 | 0 | 0 | 0 |
| 73 | 1 | 1 | 0 | 0 | 0 | 0 | 0 | 1 |
| 74 | 1 | 0 | 0 | 0 | 0 | 0 | 0 | 0 |
| 75 | 1 | 0 | 1 | 1 | 1 | 1 | 0 | 1 |
| 76 | 1 | 1 | 1 | 1 | 1 | 0 | 0 | 0 |
| 77 | 1 | 1 | 0 | 0 | 0 | 1 | 1 | 0 |
| 78 | 1 | 1 | 1 | 0 | 1 | 0 | 0 | 1 |
| 79 | 0 | 0 | 0 | 0 | 0 | 0 | 0 | 0 |
| 80 | 0 | 1 | 0 | 0 | 0 | 1 | 1 | 0 |
| 81 | 0 | 0 | 0 | 0 | 0 | 0 | 0 | 0 |
| 82 | 0 | 0 | 0 | 0 | 0 | 0 | 0 | 0 |
| 83 | 1 | 1 | 1 | 1 | 0 | 0 | 0 | 1 |
| 84 | 1 | 1 | 1 | 0 | 1 | 0 | 0 | 0 |
| 85 | 1 | 0 | 1 | 0 | 0 | 0 | 0 | 0 |
| 86 | 0 | 0 | 0 | 0 | 1 | 0 | 0 | 1 |
| 87 | 0 | 0 | 0 | 0 | 0 | 0 | 0 | 0 |
| 88 | 0 | 0 | 1 | 0 | 0 | 0 | 1 | 0 |
| 89 | 1 | 1 | 0 | 0 | 0 | 0 | 0 | 0 |
| 90 | 1 | 1 | 0 | 0 | 1 | 0 | 0 | 0 |
| 91 | 1 | 0 | 0 | 0 | 0 | 0 | 1 | 1 |
| 92 | 1 | 0 | 1 | 1 | 1 | 0 | 0 | 0 |
| 93 | 0 | 0 | 0 | 0 | 0 | 0 | 0 | 0 |
| 94 | 1 | 1 | 1 | 1 | 1 | 0 | 1 | 0 |
| 95 | 1 | 1 | 0 | 1 | 0 | 0 | 1 | 1 |
| 96 | 1 | 1 | 0 | 0 | 1 | 0 | 0 | 1 |
| 97 | 1 | 1 | 1 | 1 | 1 | 1 | 1 | 1 |
| 98 | 1 | 1 | 0 | 1 | 1 | 0 | 1 | 1 |
| 99 | 1 | 1 | 1 | 0 | 0 | 0 | 1 | 1 |
| 100 | 0 | 1 | 1 | 1 | 1 | 0 | 0 | 0 |

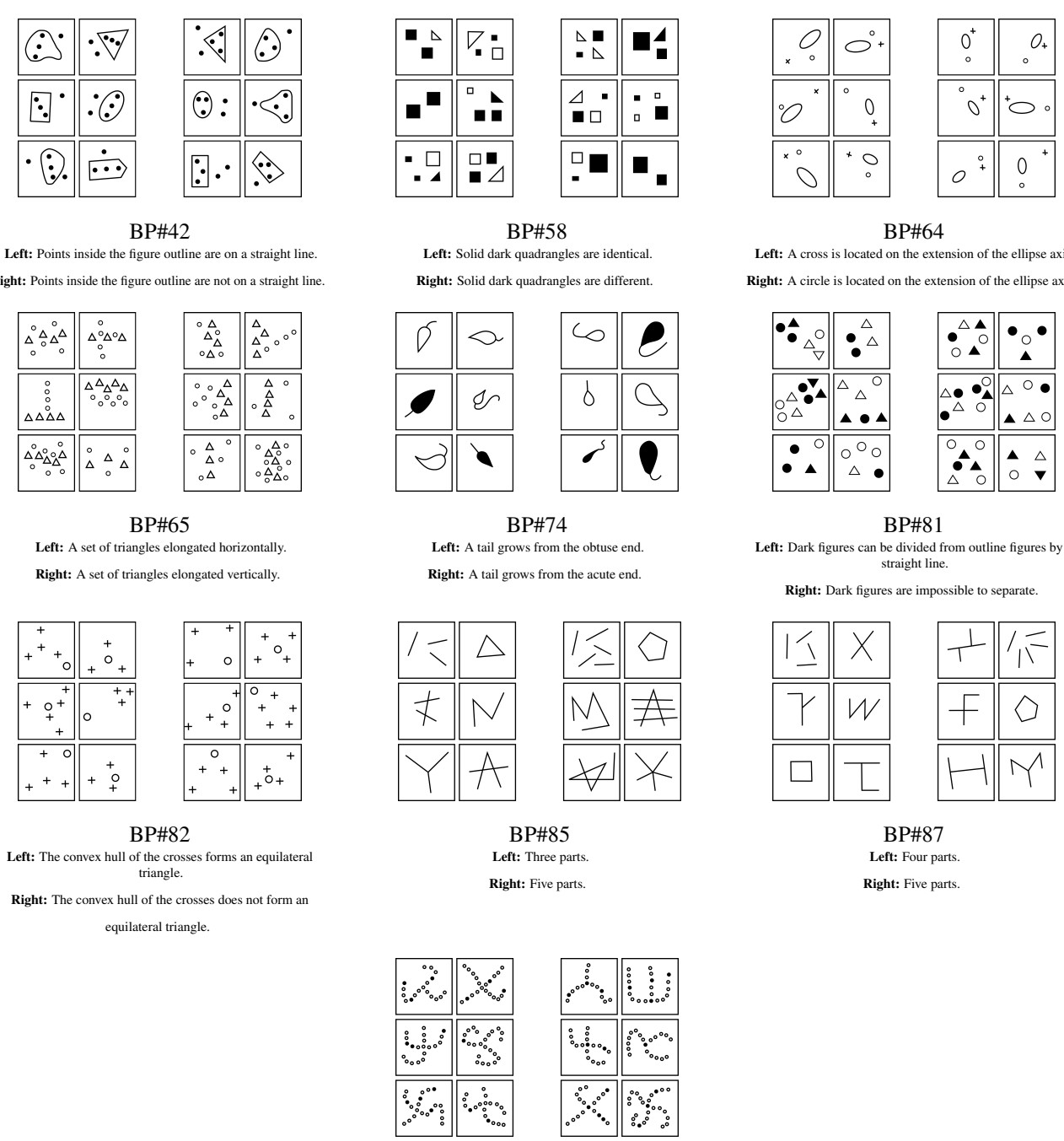

**BP#42**

**Left:** Points inside the figure outline are on a straight line.

**Right:** Points inside the figure outline are not on a straight line.

**BP#58**

**Left:** Solid dark quadrangles are identical.

**Right:** Solid dark quadrangles are different.

**BP#64**

**Left:** A cross is located on the extension of the ellipse axis.

**Right:** A circle is located on the extension of the ellipse axis.

**BP#65**

**Left:** A set of triangles elongated horizontally.

**Right:** A set of triangles elongated vertically.

**BP#74**

**Left:** A tail grows from the obtuse end.

**Right:** A tail grows from the acute end.

**BP#81**

**Left:** Dark figures can be divided from outline figures by a straight line.

**Right:** Dark figures are impossible to separate.

**BP#82**

**Left:** The convex hull of the crosses forms an equilateral triangle.

**Right:** The convex hull of the crosses does not form an equilateral triangle.

**BP#85**

**Left:** Three parts.

**Right:** Five parts.

**BP#87**

**Left:** Four parts.

**Right:** Five parts.

**BP#93**

**Left:** Branches at outlined circle.

**Right:** Branches at solid dark circle.

Figure 11: **Most Challenging Bongard Problems.** Ten Bongard Problems (BPs) that were not solved by any model across our evaluations. High-resolution images adapted from (Depeweg et al., 2024).

