# OpenReview forum: "Bongard in Wonderland: Visual Puzzles that Still Make AI Go Mad?"
_ICML.cc/2025/Conference — ICML 2025 poster_

### Official Review · Reviewer_5FyS · 2025-03-09

**Overall Recommendation:** 1

**Summary:**

This paper presents a case study of utilizing VLMs for solving Bongard problems, and identifies that it remains challenging for VLMs to reason some basic concepts in Bongard problems. The authors also conduct a comparison between VLMs' and human's reasoning abilities on Bongard problems.

**Claims And Evidence:**

The authors mainly made four claims: 1) Evaluation of VLMs on identifying underlying rules. 2) Comparisons of VLMs to Human's reasoning ability. 3) Exploration of the models’ pattern recognition abilities. 4) Examination of the ability of generating hypotheses.

Those claims are mainly associated with the experimental evaluation on the Bongard problems. The authors did conduct those evaluations as shown in Sec. 4.

**Essential References Not Discussed:**

Many existing solution models for Bongard problems are not discussed or compared in the experimental validaiton. E.g.,
[1] Take A Step Back: Rethinking the Two Stages in Visual Reasoning, ECCV, 2024.
[2] Neural Prediction Errors enable Analogical Visual Reasoning in Human Standard Intelligence Tests, IMCL, 2023.

**Experimental Designs Or Analyses:**

The authors mainly evaluate existing VLMs but no proposed one. Also, no other existing solution models for Bongard problems are compared in the experiments.

**Methods And Evaluation Criteria:**

The authors mainly made use of existing VLMs for the evaluation and did not propose any new method. The evaluation critera are clear and suitable.

**Other Comments Or Suggestions:**

The authors may add some theoretical analysis in this paper to boost the technical aspects of this paper.

**Other Strengths And Weaknesses:**

Strength: The paper is well written with clear logic flow and comprehensive evaluation on VLMs.
Weakness: 1) This paper did not propose any new methods. The novelty of this paper is very limited. 2) The key finding, the incapability of VLMs in solving abstract reasoning problems, including Bongard problems, have been discussed in literature. This greatly limits the significance of this paper.

**Questions For Authors:**

1. In case I miss any novel design in this paper, can the authors justify the novel contributions of this paper?
2. Can the authors provide more evaluation results of other existing models, in comparisons to the results of VLMs and human's? Or, can the authors justify why other existing models are not chosen for comparison?
3. Can the authors provide more theoretical insights of the proposed method?

**Relation To Broader Scientific Literature:**

The main contribution of this paper is to unveil that current VLMs are not able well solve Bongard problems. But the authors did not propose any new approaches to better solve the problems. In literature, there are many existing solution models to solve the Bongard problems. But the authors did not compare with those methods, which will greatly limits the significance of the any discussions or conclusions drawn from the results.

**Theoretical Claims:**

There are no theoretical claims. The authors mainly conducted an empirical study of existing VLMs on Bongard problems.

---

> ### Author Rebuttal · Authors · 2025-04-01
>
> Thank you for taking the time to review our work and for considering it well written. We hope that the following responses will also convince you of the strength and value of its contributions.
>
> **(W1, Q1 - No new method)**
>
> We respectfully disagree with the reviewer’s assessment regarding the lack of novelty, and note the other reviewers did not raise any concerns on novelty (with reviewer fqym explicitly stating that "this work appears novel"). While we do not propose a new technique to solve Bongard problems directly (an ambitious and still unsolved challenge) we introduce a detailed typology of model behaviours in such visual tasks, revealing strengths and limitations of current VLMs, helping us to understand their learned knowledge and abilities to reason.
>
> Our contributions include:
> - Introducing various prompt configurations (open-ended and multiple-choice) to explore how different prompt strategies affect VLMs’ performance on Bongard problems.
> - Presenting two novel problem setups (Task 2 and Task 3) designed to analyze both the perceptual and the robustness aspects of VLMs.
> - Conducting a human study involving 20 participants, which allows us to compare human performance with that of VLMs and thereby uncover systematic discrepancies.
> - Identifying 10 Bongard problems that no deployed model in our study could solve in any task setup, highlighting clear gaps for future research.
> - Demonstrating a substantial discrepancy between correctly identified rules in Task 1 and those that are correctly applied in Task 2, which unveals inconsitant model behaviour as the VLM can corretly find a rule but not use it reliably.
>
> By documenting and analyzing these insights, we aim to inform the broader ML community about current limitations and future directions in solving Bongard problems. We believe this approach is in line with the guidelines’ emphasis on supporting new tasks, metrics, and problem framings, even when no novel algorithmic methods are introduced. We believe this set of contributions constitutes meaningful progress that the ICML community can build upon.
>
> **(W2 - Key findings have been discussed in literature already)**
>
>  Even though there exists other works that investigate the shortcomings in abstract visual reasoning in VLMs, we think that several new insights can be drawn from our work, see the listed contributions above. For example our work identified a gap between problem solving (Task 1) and perception or rather applying rules correctly (Task 2) that is, to our knowledege, not yet discussed in the literature.
>
> **(Q2 - Compare to other methods)**
>
> We appreciate your interest in seeing broader evaluation results. However, our primary aim is not to identify the single best method for solving Bongard problems, but rather to reveal critical limitations and uncover how current VLMs, which can be opaque in their reasoning, perform on these tasks. We aspire to highlight systematic insights that can guide future improvements in model design and evaluation.
>
> That said, we can discuss other methods that aimed at solving Bongard problems in more detail in related work. However they usually have a very different setup to VLMs and our aim is not to propose a method for solving Bongard problems, instead we use them as a diagnostic dataset to understand and analyse VLMs in more detail.
>
> In that light we also consider the proposed references. Both present interesting approaches; however, they focus on the Bongard-LOGO dataset rather than the original Bongard problems considered in our study. While the approach in [1] could, in principle, be adapted to the original Bongard problems in an open-ended setting, [2] is specifically designed for classification tasks, making it difficult to apply directly to the open-ended nature of the original Bongard problems. Moreover, our primary focus is on evaluating the capabilities of VLMs independently of dedicated reasoning architectures, as their underlying mechanisms differ significantly. Nonetheless, we have included both works in our related work section to acknowledge their contributions.
>
> **(Q3 - more theoretical insights)**
>
> We are a bit puzzled by this remark. As we have highlighted above the contribution of this work is not to propose a novel AI method, but rather introduce and analyze a valuable dataset and typology of model behaviour for investigating this dataset in the context of VLMs. Can the reviewer clarify what they are referring to with "theoretical insights of the method"?

---

### Official Review · Reviewer_uWew · 2025-03-13

**Overall Recommendation:** 4

**Summary:**

The paper benchmarks existing vision-language models using Bongard problems (BPs). It also performs a human evaluation for comparison. The paper tests not only whether a model can solve a given BP or not, but also whether the main concept in the BP can be recognized in the individual images in the BP, and whether the model can generate the correct solution when it is asked to generate a set of candidate hypotheses. The results are surprising, and show, for example, that models not only significantly underperform humans, but they also do not seem to correctly perceive the individual images even in cases where they correctly solve a problem.

**Claims And Evidence:**

Yes.

**Essential References Not Discussed:**

N/A

**Experimental Designs Or Analyses:**

Yes.

**Methods And Evaluation Criteria:**

Yes. The paper performs an evaluation of existing, pre-trained models on tasks derived from Bongard problems to compare to human baseline and to assess the performance consistency between the different tasks.

**Other Comments Or Suggestions:**

Footnote 3: “Exception: In Task 2 o1 was prompted once.” Why was that?

Even though the performance of models is fairly low, as the problems are publicly available (and have been for a long time) is there any chance of contamination affecting the results? It seems that even the inconsistency regarding Task 1 vs Task 2 might be explainable to some degree through contamination as well (with Bongard problems - or similar types of problems - seen during training enhancing the ability to generate a shortcut answer without truly perceiving details of the images)?

**Other Strengths And Weaknesses:**

The paper is very clear, well structured, and easy to follow, and it presents lots of interesting insights.

The comparison between Task 1 (solving BPs) and Task 2 (detecting a BP’s underlying concepts in individual images) is quite nice and the results are revealing. And it seems in line with the findings for Task 3. It is nice that it also highlights a danger in reading too much into the ability to solve any given problem (using the Task 1 setting).

Contrarily, it seems that the danger also applies to human evaluations. And it suggests a human evaluation for Task 2 would be very helpful to complete the picture and would significantly strengthen this study. Have the authors considered this?

**Questions For Authors:**

The presence of multiple panels in a single image could be difficult for existing models to process simply because information is highly local as a result. And it seems that in some cases this could make the resolution in which a given model perceives the local panels too small to perceive details (some models downsample any given images to a fixed resolution, such as 224x224, before processing them).

Could this be a confounding factor (especially for Task 2 and the results of Task 1 vs Task 2)? One way to help ensure that this is not the reason for the observed failure cases would be to present the panels separately, each in an individual image, rather than within a single. Have the authors considered this?

**Relation To Broader Scientific Literature:**

The paper provides an overview of existing work on Bongard problems and similar tasks to evaluate AI models. The references seem fairly comprehensive as far as I can tell. The study in this work differs in substantial ways from existing similar studies.

**Theoretical Claims:**

N/A

This is an empirical study.

---

> ### Author Rebuttal · Authors · 2025-04-01
>
> Thanks for your detailed response and the constructive feedback! Below we address your concerns.
>
> **(W1 - Human study for Task 2)**
>
> We agree that analyzing human performance in Task 1 alongside Task 2 would be an interesting future direction in a different setting with higher conceptual ambiguity and novel concepts. However, our primary focus in this paper was identifying the VLMs' perception errors in relation to their performance in Task 1.
>
> We hypothesize that humans would excel in Task 2, as previous literature has identified concept recognition, spatial reasoning, and relational abstraction as fundamental aspects of human cognition [1,2]. Given that Bongard problems involve discriminative rules, we argue that human performance should remain robust in this task.
>
> While rule verification tasks can, in principle, be challenging for humans in cases of rule ambiguity or when unfamiliar concepts are introduced, this is unlikely in the case of Bongard problems due to the discriminative nature of the rules. For example, in BP#16, asking a human whether a spiral turns clockwise or counter-clockwise (starting from the center) should be straightforward, as the rule is well-defined.
>
> [1] Lake BM, Ullman TD, Tenenbaum JB, Gershman SJ. Building machines that learn and think like people. Behavioral and Brain Sciences. 2017
>
> [2] Gentner, D. (2003). "Why we’re so smart." Language in Mind: Advances in the Study of Language and Thought
>
> **(C1 - Footnote o1 prompting)**
>
> The o1 evaluation is quite expensive, therefore we decided to evaluate only once for the concept detection experiment, as it would required 2400 more requests. If the reviewer thinks, it would be valuable to have the additional trials for o1 as well, we can still retrieve these results.
>
> **(C2 - BPs in training Set)**
>
> Unfortuantly, the opaque nature of the training processes, particularly for models developed by large corporations with proprietary datasets, makes it impossible to determine whether the models have actually been exposed to the BPs during training. However, even if such examples were present, the low overall performance of the models indicates that they have certainly not fully comprehended them. Your suggestion that the models might have learned shortcuts, could be one explanation why many Bongard problems solved in Task 1 remain unsolved in Task 2. It would be interesting future work to investigate this phenomenon on non-public test sets. Overall, this is an interesting hypothesis, and we included it in our discussion.
>
>  **(Q1 - Representation of BPs (single images))**
>
> Interesting suggestion, we have investigated this with some of the models (c.f. table below). We see that the performance in Task 1 is comparable but more interestingly also the behaviour between Task 1 and Task 2 stays similar. This suggests that the image representation alone cannot be the reason for this discrepancy. We included these findings in the final paper.
>
> |  | GPT-4o | Claude 3.5 |
> | -------- | -------- | -------- |
> | Solved BPs Original Setup     | 25     | 31
> | Solves BPs Single Images | 25 | 33|
>
> Results analogously to Figure 5:
>
> | | T1 w/o T2 | T1 $\cap$ T2 | T2 w/o T1
> | -------- | -------- | -------- | -------- |
> | GPT-4o (orig)     |13 | 11 | 13 |
> | GPT-4o (single imgs)     |12 | 12 | 12 |
> | Claude (orig)     |20 | 11 | 11 |
> | Claude (single imgs)     |20 | 12 | 10 |

---

### Official Review · Reviewer_R3ti · 2025-03-13

**Overall Recommendation:** 3

**Summary:**

This paper explores the performance of VLMs on Bongard problems. To test the abstract reasoning ability of VLMs, three different types of tasks are proposed: (1) open-ended solving of Bongard problems, (2) detection of specific concepts, and (3) formulation of hypotheses. Task 1 is to summarize the rules of the left and right panels of Bongard problems, or to select the rules from some options. Task 2 requires VLMs to identify whether a certain image follows a certain rule or concept. Task 3 tests the model's ability to generate Bongard problem rules. The performance on the above tasks can evaluate the robustness and reasoning ability of VLMs. The experimental results show that there is still a large gap between the reasoning ability of VLMs and humans. This work provides valuable insights for evaluating the concept learning and abstract reasoning abilities of current VLMs.

**Claims And Evidence:**

The claims made in this paper are clear and supported by its experiments.

**Essential References Not Discussed:**

The related works that are essential have been discussed in this paper.

**Experimental Designs Or Analyses:**

This paper is reasonable and effective in experimental design and analysis.

**Methods And Evaluation Criteria:**

The datasets and evaluation criteria of this paper make sense for the problem.

**Other Comments Or Suggestions:**

As important forms of abstract visual reasoning tasks, the authors can further incorporate raven matrices or odd-one-out problems into the test. These problems can also be transformed into open-ended forms, like Bongard problems, by describing the rules to complete the task instead of choosing the result from the options.

**Other Strengths And Weaknesses:**

Strengths

This paper refines traditional Bongard problems and proposes three different tasks, which respectively verify the rule induction, concept learning and rule imagination abilities of VLMs. The performance of the above tasks can reveal different dimensions of VLMs' abstract visual reasoning ability. Therefore, this paper provides some insights into the reasoning ability of commonly used VLMs.

Weaknesses

The main concern is about the data collection process. In this work, only 100 Bongard problems from existing work are selected as test data. Could the authors provide a detailed introduction and discussion on the data collection process? For example, for what considerations and what criteria were used to select these Bongard problems. Why not use larger Bongard problem datasets, e.g., CVR [1] and SVRT [2]. CVR and SVRT include a large number of program-generated Bongard problems, which should have annotations of rule descriptions and concept labels for the panel images.

[1] Zerroug, Aimen, et al. A benchmark for compositional visual reasoning.
[2] Fleuret, François, et al. Comparing machines and humans on a visual categorization test.

**Questions For Authors:**

I notice in Figure 4 that the accuracy of the human answers to the questions varied quite a bit (e.g., from 0% to 100% in the "same" rule). I wonder why the participants have such a large variance in their results. Is it due to the difference of participants' abstract reasoning ability, or because the participants do not understand the form of the Bongard problems?

The bar chart on the right of Figure 4 does not show the black area representing "solved", so what does the "solved" black block of the legend mean?

**Relation To Broader Scientific Literature:**

The main contribution of this paper is to verify abstract visual reasoning ability of current VLMs on Bongard problems. Previous work [1] verified abstract visual reasoning ability of VLMs on non-open-ended problems like Raven matrices and odd-one-out problems, which is different from this work. This paper analyzed the concept learning ability in Bongrad problems, which is not covered by previous works.
[1] Cao, Xu, et al. What is the visual cognition gap between humans and multimodal llms?

**Theoretical Claims:**

This paper does not involve theoretical claims and proofs.

---

> ### Author Rebuttal · Authors · 2025-04-01
>
> Thank you for your valuable feedback. We respond to your points in detail below.
>
> **(W1 - Selection of test data)**
>
> We chose to work with the original Bongard problems introduced in [1], as they were specifically designed to test pattern recognition capabilities in machines, yet they remain unsolved by current AI systems. These problems were carefully crafted and, in our view, already present a significant challenge on their own. Importantly, we find that they strike a valuable balance between simplicity (in terms of visual size and structure) and conceptual difficulty, a point supported by our accompanying human study.
>
> We agree that datasets such as CVR and SVRT are also interesting, as they share some conceptual elements with Bongard problems, including shape variation, counting, insideness, and contact. Exploring how VLMs perform on these datasets would be an exciting direction for future work.
>
> [1] Bongard, M. (1970). Pattern recognition. New York: Spartan Books
>
> **(S1 - Add abstract visual reasoning tasks)**
>
> Thank you for the thoughtful suggestion. We agree that Raven’s Progressive Matrices and odd-one-out tasks are important forms of abstract visual reasoning, and we appreciate the idea of transforming them into open-ended formats. However, as motivated in the paper, we believe that Bongard problems already provide a strong and sufficiently challenging foundation for evaluating the kinds of open-ended, concept-based reasoning we are interested in.
> In fact, many of the underlying visual concepts of Raven and odd-one-out (e.g., sameness, symmetry, numerical relations) are already well-represented in the Bongard problems.
>
> That said, we agree that adapting other reasoning formats into an open-ended, explanation-driven setup as the reviewer suggests could be an exciting direction. While not necessary for the scope of the current study, we see this as promising future work, especially for expanding evaluation diversity and probing generalization across reasoning formats.
>
> **(Q1 - same BPs variance)**
>
> The "same" category includes only seven Bongard problems (see Table 4), which naturally increases the variance of participants’ results and makes it difficult to draw strong conclusions. It is possible that some individuals find the concept of "sameness" more intuitive than others, though further study would be required to explore this systematically.
>
> In general, high variance across participants may stem from differing levels of task comprehension. For instance, the lowest-scoring participant - who did not solve any Bongard problems in the ‘same’ category - often formulated a rule that applied only to the left side of a problem while dismissing the right side as simply “not following that rule.” In #BP98, they labeled the classes as “triangle shapes” and “non-triangle shapes,” overlooking the crucial contrast between “triangles” and “quadrangles.” Such answers suggest they may not have realized that merely stating the inverse of a rule is insufficient to fully characterize a Bongard problem. Similarly, this participant used relative terms like “more circle shapes” vs. “fewer circle shapes,” highlighting a misunderstanding of the need for clearly defined classification rules independent of context.
>
> **(Q2 - black area in legend)**
>
> The legend entry for "solved" is meaning filled rectangle (red or blue) rather than black filling. We see that this can be misleading. To improve clarity, we consider updating it to a half-red, half-blue filled rectangle. Does the reviewer think that this would improve clarity?

---

> > ### Comment · Reviewer_R3ti · 2025-04-05
> >
> > I thank the authors for the detailed responses, which have solved my questions. I think it is reasonable to change the black rectangle to a half-red-half-blue one. Now I can understand the legend well. I would like to keep my original score.

---

> > > ### Author Response · Authors · 2025-04-08
> > >
> > > We thank the reviewer for their response and the feedback, we updated the figure accordingly. We are happy to hear that we were able to address all the questions. In light of this, we kindly ask the reviewer if they could reflect their score and consider raising it.

---

### Official Review · Reviewer_fqym · 2025-03-16

**Overall Recommendation:** 3

**Summary:**

The paper evaluates current VLMs on Bongard Problems. Each Bongard Problem (BP) consists of 12 images divided into two sides — the left side and the right side — each side containing 6 images. The images on each side are characterized by a rule not shared by the other side. The aim of the problem solver is to identify the rule pair that applies to each problem — via a text description of the rules. The paper develops 3 tasks: (1) Task 1 focuses on the open-ended generation of the rule given the problem image. (2) Task 2 focuses on providing the ground truth rule pair in the context and then asking the model to classify each of the 12 images into one of the two given rules. (3) Focuses on asking for multiple rule-pair hypotheses for each problem, unlike Task 1 where the model needs to generate the correct rule pair in a single go. The paper also evaluates human performance on Task 1. The paper then reports the results of these tasks for various available VLMs and infers several interesting conclusions.

**Claims And Evidence:**

Yes. The paper claims that there is a significant gap between VLMs and human performance and this is made quite clear from the experiments.

**Essential References Not Discussed:**

I am not aware of such references.

**Experimental Designs Or Analyses:**

Yes, the experimental design is sound. I will ask any doubts in the "Questions for Authors" below.

**Methods And Evaluation Criteria:**

Yes.

**Other Comments Or Suggestions:**

See Cons and Questions.

**Other Strengths And Weaknesses:**

## Pros

1. Interesting findings and questions raised for future work.
2. The paper is well-written. The way the metrics are reported is clear. The experiment section provides a large number of new and interesting insights.
3. Identifying the most difficult problems not solved by any model could be a very useful resource for the community.

## Cons

I think there needs to be a more in-depth discussion of the differences between this paper and the paper of Malkinski et al. 2024. I don’t mean to imply that there aren’t differences but rather that the reader should be able to see them more clearly. Currently, the authors say “While they (Malkinski et al. 2024) provide meaningful insights, they only consider a classification setting for the evaluation and do not investigate the model behavior in more depth.”, but this seems not specific enough. For instance: Malkinski et al.’s paper also does direct generation (akin to Task 1, if I am not mistaken).

It may be nice to mention which conclusions could not have been arrived at with previous work’s experiments and what are their implications for the future. The results are very interesting as standalone statements, but I am not sure if I have been able to grasp the main (and novel) takeaway message about VLMs from the paper.

**Questions For Authors:**

1. Are the solutions to Bongard problems public and present in pre-training datasets of the VLMs?
2. Were multiple outputs sampled from the models to ascertain that a task is solved (i.e., similar to pass@$k$)?
3. Was human study considered for Task 2?
4. Task 2 is about exploring perception. What is the rationale for choosing this specific form of perception task (e.g., why not ask in an open-ended manner about the concepts present in each image)? I am not sure if I see a clear direct connection with the conclusion noted later “perception is a key issue for not identifying the correct rules of BPs”?
5. Authors say in L355: “surprising gap between recognizing correct classifications and effectively applying that knowledge in problem-solving”. However, a solver might try to “guess” a rule even if it seems to apply to the majority of the images (if not all) on each side. Wondering about the author’s thoughts on this.
6. Is there a technical value associated with the contents of Fig. 1 (e.g., the person’s face)?

**Relation To Broader Scientific Literature:**

The paper is about evaluating the limitations of VLMs in how well they can understand abstract concepts. For this, the paper looks at the Bongard problems. Among the works about VLMs that look at Bongard problems, this work appears novel. I request further clarity from the authors regarding this via the Cons listed below.

**Theoretical Claims:**

Not applicable.

---

> ### Author Rebuttal · Authors · 2025-04-01
>
> Thank you for your detailed feedback and questions, we address them below.
>
> **(W1 - Differences to Malkinski et al.)**
>
> Malkinski et al. (2024) also evaluated VLMs on Bongard problems, concentrating on open-ended and classification-based settings. Our work shares their open-ended focus but goes further by adding two additional tasks (Tasks 2 and 3) to examine specific abilities we hypothesize are vital for solving Bongard problems. While Malkinski et al. compare synthetic Bongard problems to their real-world variant (Bongard-RWR), we target the original Bongard set, pinpointing especially challenging cases and identifying concept-detection inconsistencies. We also compare model output directly with human performance, offering more granular insights into how VLMs reason about these complex tasks. We have added this discussion to the related work section for greater clarity.
>
> **(W2 - Takeaway messages)**
>
> In the following we outline the key takeaway messages of our work.
>
> - We introduce a typology of model behaviors for visual tasks, revealing both the strengths and weaknesses of current VLMs.
> - Models often fail even with multiple-choice answers, indicating challenges not only in discovering correct rules, but also in recognizing when they have the right one (Task 3).
> - We uncover a pronounced gap (Fig. 5) between solving Bongard problems (Task 1) and consistently identifying their relevant concepts (Task 2), highlighting an underexplored interplay between abstract reasoning and perception.
> - We identify 10 Bongard problems that no model solved under any task condition (Fig. 11), offering a strong basis for future benchmarking.
> - Our human study shows key differences between VLMs and people: participants collectively solved 95 Bongard problems (averaging 42 each, with best participant at 68), whereas all models combined solved only 66.
>
> **(Q1)**
>
> Unfortunately, the training data for both closed-source and open-source VLMs is not publicly available, so we cannot determine this with certainty. However, given that these large models are typically trained on massive corpora that include a substantial portion of publicly available data, it is likely that they have been exposed to Bongard problems during training - especially since the Bongard problems, along with their solutions, are publicly accessible (e.g., https://www.oebp.org/welcome.php). It would be interesting future work to design versions of BPs that are private or could get generated automatically to avoid the public exposure.
>
> **(Q2)**
>
> Yes, models were sampled three times. Tasks 1 and 2 were considered correct if answers/images were correct in at least 2 of 3 attempts. For Task 3, we sampled once and checked if a correct hypothesis was among the 20 proposed.
>
> **(Q3)**
>
> Please refer to W1 of reviewer uWew for more details.
>
> **(Q4)**
>
> Our goal in Task 2 is to examine whether models that successfully identify a discriminative rule (Task 1) also apply that rule consistently to each individual image. We hypothesize that a truly correct solution should translate into accurate classification of all images in a Bongard problem. However, our findings show that a model can solve a Bongard problem in Task 1 yet still classify certain images incorrectly in Task 2. For instance, GPT-4o solves BP#15 - distinguishing open from closed shapes - in all three Task 1 trials but fails to label an open triangle as open in Task 2. This indicates that even when VLMs manage to solve a Bongard problem, they may not reliably apply their “understood” rule at the level of individual images. While asking open-ended questions about each image (rather than providing ground-truth concepts) is an alternative approach, our intention here is to assess the consistency between rule discovery and rule application under controlled conditions.
>
> **(Q5)**
>
> We agree that a model may sometimes approximate the correct solution without consistently applying it across all images. However, this is not necessarily the behavior we seek in VLMs, particularly in applications that demand reliability and robustness. This highlights the importance of considering the perceptual aspect together with the reasoning abilities when evaluating VLMs.
>
> That said, our comment was pointing to a complementary finding: that some ground-truth concepts underlying Bongard problems are well-detected by the models, even though they do not emerge as the rule for solving the BP in the general task. This may stem from confounding visual factors or the limited prominence of some ground-truth concepts. We find this to be an interesting observation, and think that it opens up promising directions for future work.
>
> **(Q6)**
>
> The figure is rather illustrative, however we intended to give an impression of different possible diagrams occuring in Bongard problems (displayed on the cards).

---

> > ### Comment · Reviewer_fqym · 2025-04-06
> >
> > Thank you for the responses!  I think the results are interesting as individual nuggets and, at the same time, feel that the paper’s take-away message is a bit blurry and not tied together into a clear message about VLMs and what the community can do about it in the future.
> >
> > > Models often fail even with multiple-choice answers, indicating challenges not only in discovering correct rules but also in recognizing when they have the right one.
> > >
> >
> > > … even when VLMs manage to solve a Bongard problem, they may not reliably apply their “understood” rule at the level of individual images.
> > >
> >
> > I think there is something interesting going on here, and would be nice to have a more in-depth exploration of these.
> >
> > I’m fine accepting the paper, but I can understand if it’s not accepted in its current state. I will maintain the score for now.

---

> > > ### Author Response · Authors · 2025-04-08
> > >
> > > We thank the reviewer for continuing the interesting discussion. We appreciate that they find the individual results compelling, yet seek a clearer, more cohesive message. Below, we restate our core contributions and propose future directions for the community:
> > >
> > > **1. Bongard Problems (BPs) still fundamentally challenge modern VLMs.**
> > > - Even though VLMs can sometimes propose (or choose) correct rules for an entire BP (Task 1), they frequently fail to apply those rules consistently at the individual image level (Task 2).
> > > - This points to gaps not only in abstract rule-formulation but also in perceptual grounding and reliable rule application.
> > > - We observe a pronounced lack of spatial reasoning (one of the lowest-scoring BP categories).
> > > - A considerable performance gap persists between humans and VLMs on BPs.
> > >
> > > **2. VLMs’ uncertainty about “having the right rule” is a persisting bottleneck.**
> > > - Our results show that, in multiple-choice formats, models still often pick incorrect answers, indicating they struggle to self-check their own hypotheses, even when the solution set is provided.
> > > - This suggests that the challenge is twofold: (a) discovering a discriminative rule and (b) recognizing when the discovered rule actually fits the data.
> > >
> > > **3. Concrete takeaways and potential improvements for the community:**
> > > - Multi-stage reasoning pipelines: Recent hints from subsets of our experiments (Type II, Type III behaviors) suggest models can encode aspects of the correct rule but fail to integrate them end-to-end. A structured approach that explicitly ties high-level rule discovery to lower-level, image-by-image verification may help.
> > > - Perceptual consistency: Many BPs rely on basic spatial or geometric features (e.g., "left" vs. "right", "inside" vs. "outside", or counting shapes). Our results confirm that even advanced VLMs stumble on fundamental perception tasks, underscoring how perception-based reliability remains a major pain point.
> > > - “Self-monitoring”: The shortfall in multiple-choice answering suggests that systematically evaluating and revising potential rules (rather than returning a single best guess) might nudge models toward more accurate, self-consistent outputs.
> > > - Mechanistic Interpretability: To better understand model behavior, future work could explore the internal representations of VLMs to determine whether they truly integrate visual concepts with abstract rule reasoning.
> > >
> > > **4. Broader significance of our findings:**
> > > - The community can leverage our systematic failure modes, especially Type II (correct rule hypothesized, but not applied when solving the BP) and Type III (BP solved, but individual-image classification is inconsistent), to pinpoint areas of improvement for next-generation multimodal architectures.
> > > - Going forward, creating novel Bongard-like tasks or expanding the original set with private or auto-generated puzzles could mitigate training-data contamination and further stress-test how (and whether) models grasp visual abstractions.
> > >
> > > In summary, our main goal of the paper is to demonstrate that BPs are not just "nice-to-have" puzzles but diagnostic tasks spotlighting surprisingly fundamental gaps in VLM performance. Above we have detailed our take-away messages towards VLM development. We hope these clarifications show a cohesive narrative, namely, that BPs reveal both a need and a path for improving the reliability, interpretability, and fine-grained reasoning processes of VLMs.
> > >
> > > We have expanded our paper's discussion section (section 5) to discuss the above mentioned points in more depth and extended our section on future work.
> > >
> > > We hope this discussion clarifies the reviewer's concerns and would appreciate if they could reconsider your score.

---

### Decision · Program_Chairs · 2025-05-01

**Decision:**

Accept (poster)

**Comment:**

This study evaluates LLMs on classical Bongard problems, which require human-like reasoning abilities, and finds that LLM behavior diverges significantly from human reasoning patterns. While one reviewer questioned the novelty of this work, others provided positive feedback and acknowledged its value. Given these assessments, I recommend accepting this paper.